# POKEFLEX: A REAL-WORLD DATASET OF VOLUMETRIC DEFORMABLE OBJECTS FOR ROBOTICS

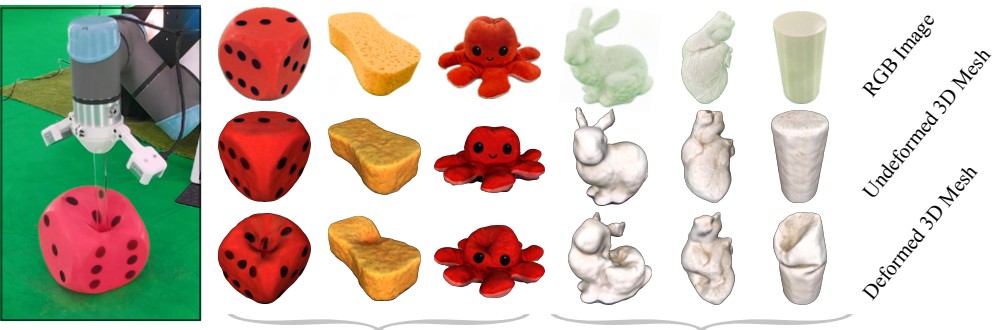

Figure 1: PokeFlex captures the deformability of various everyday and 3D-printed objects, as illustrated by the poking manipulator on the **Left**. On the **Right**, the **Top Row** contains segmented RGB images of selected objects. The **Middle Row** shows reconstructed objects in an undeformed state. The **Bottom Row** provides reconstructed 3D-textured meshes of deformed objects.

## ABSTRACT

Data-driven methods have shown great potential in solving challenging manipulation tasks, however, their application in the domain of deformable objects has been constrained, in part, by the lack of data. To address this, we propose PokeFlex, a dataset featuring real-world paired and annotated multimodal data that includes 3D textured meshes, point clouds, RGB images, and depth maps. Such data can be leveraged for several downstream tasks such as online 3D mesh reconstruction, and it can potentially enable underexplored applications such as the real-world deployment of traditional control methods based on mesh simulations. To deal with the challenges posed by real-world 3D mesh reconstruction, we leverage a professional volumetric capture system that allows complete 360° reconstruction. PokeFlex consists of 17 deformable objects with varying stiffness and shapes. Deformations are generated by dropping objects onto a flat surface or by poking the objects with a robot arm. Interaction forces and torques are also reported for the latter case. Using different data modalities, we demonstrated a use case for our dataset in online 3D mesh reconstruction. We refer the reader to our website[1] or the password protected supplementary material[2] for further demos and examples.

## 1 INTRODUCTION

Data-driven methods have recently demonstrated promising results in deformable object manipulation, significantly advancing automation in industries such as healthcare, food processing, and manufacturing (Bartsch et al., 2024; Deng et al., 2024; Avigal et al., 2022; Yan et al., 2021). To further advance research in this area, the development of high-quality datasets is essential. Such datasets are crucial for training manipulation policies, estimating material parameters, and training 3D mesh reconstruction models. The latter, in particular, plays a vital role in facilitating the close-loop execution of control methods based on mesh simulations (Duenser et al., 2018). In light of

---

[1] https://anonymized-pokeflex-dataset.github.io/

[2] https://drive.google.com/drive/folders/1d8iNoJZ0dUVlzP6XxP7xwGPhdVtwQ7du
Password: P0keFlex-ICLR2025-Dataset

Table 1: Dataset overview (per object, per sequence).

| Sequence Data | Poking | Dropping |
|---|:---:|:---:|
| 3D textured deformed mesh model | ✓ | ✓ |
| RGB images from two Volucam cameras (cameras from the MVS) | ✓ | ✓ |
| RGB-D images from two RealSense D405 sensors (eye-in-hand mounted) | ✓ | |
| RGB-D images from two Azure Kinect sensors (eye-to-hand mounted) | ✓ | |
| Estimated 3D contact forces and torques | ✓ | |
| End-effector poses | ✓ | |
| **Camera and Object Data** | | |
| Camera intrinsic and extrinsic parameters | ✓ | |
| 3D textured template mesh model | ✓ | |
| Open-source print files to reproduce the 3D printed objects | ✓ | |

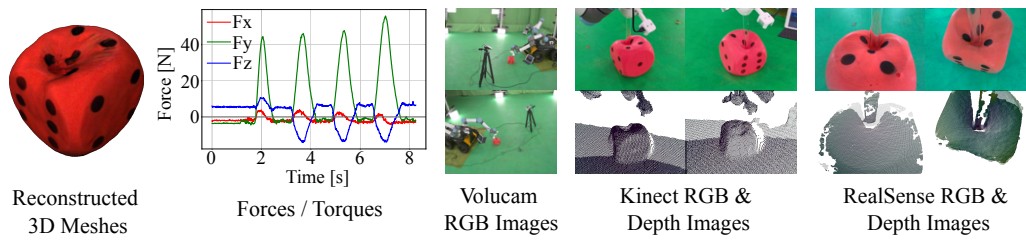

| Reconstructed 3D Meshes | Forces / Torques | Volucam RGB Images | Kinect RGB & Depth Images | RealSense RGB & Depth Images |

Figure 2: Samples of different data modalities provided by the PokeFlex dataset.

these needs, the objective of this work is to create a reproducible, diverse, and high-quality dataset for deformable volumetric objects that is grounded in real-world data.

Current state-of-the-art simulation methods can be an attractive alternative to collect such datasets as they provide easy access to privileged information such as deformed mesh configurations and contact forces (Tripicchio et al., 2024; Huang et al., 2022; Macklin, 2022; Qiao et al., 2021; Todorov et al., 2012; Faure et al., 2012). However, such simulators require careful system identification and fine-tuning to address the sim-to-real gap, which ultimately requires real-world data. Static scans rotating around the scene (Pai et al., 2001; Garcia-Camacho et al., 2022; Lu et al., 2024) or custom multi-camera systems (Chen et al., 2022) can be used to collect real-world 3D models. The former can be excessively time-consuming and is unsuitable to capture temporal dynamics. The latter requires careful synchronization and data curation, especially when using noisy lower-cost sensors.

To address these challenges, we leverage a professional multi-view volumetric capture system (MVS) that allows capturing detailed 360° mesh reconstructions of deformable objects over time (Collet et al., 2015), which we use as ground-truth meshes. We integrate a robotic manipulator with joint-torque sensing capabilities into the MVS, enabling contact force estimation and facilitating automated data collection. Moreover, to enhance reproducibility and to expand the diversity of data modalities, we also integrate and synchronize lower-cost Azure Kinect and Intel RealSense D405 RGB-D sensors into the MVS.

Our work proposes the PokeFlex dataset (Figure 1), featuring the real-world behavior of 17 deformable objects, including everyday and 3D-printed objects. Deformations are generated via controlled poking and dropping protocols. An overview of the paired, synchronized, and annotated data is presented in Table 1, and illustrated in Figure 2. We demonstrated a use case of the Poke-Flex dataset, proposing baseline models capable of ingesting PokeFlex multimodal data. We present evaluation criteria for benchmarking the results. Specifically, we train neural network models for deformed mesh reconstructions based on template meshes and various input data modalities, including images, point clouds, end-effector poses and forces. The proposed architectures are suitable for online applications, reconstructing 3D meshes at a range from 106 Hz to 215 Hz depending on the input data modality, on a desktop PC with an NVIDIA RTX 4090 GPU. The pretrained models will be available with the PokeFlex dataset.

Table 2: Feature comparison of the PokeFlex dataset with other deformable object datasets.

| | Real-world | Meshes | Point clouds | RGB images | Force torque | # of objects | # of time frames | Type of deformation |
|---|---|---|---|---|---|---|---|---|
| **PokeFlex (ours)** | ✓ | ✓ | ✓ | ✓ | ✓ | 17 | 19k | Poke, drop |
| HMDO (Xie et al., 2023) | ✓ | ✓ | | ✓ | | 12 | 2,166 | Manual[†] |
| PLUSH (Chen et al., 2022) | ✓ | | ✓ | ✓ | Force[‡] | 12 | 22.84k | Airstream |
| DOT (Li et al., 2024) | ✓ | | ✓ | ✓ | | 4 | 117k | Manual |
| Household Cloth Object Set (Garcia-Camacho et al., 2022) | ✓ | ✓[§] | | ✓ | | 27 | 67 | / |
| Defgraspsim (Huang et al., 2022) | | ✓ | | | | 34 | 1.1M | Grasp |

[†] by hand    [‡] by providing air nozzle poses    [§] for ten static scenes of the cloth objects folded

## 2 RELATED WORK

**Deformable object datasets.** Depending on the use of synthetic or real-world data, deformable object datasets can be roughly categorized into two major groups. Huang et al. (2022), for instance, evaluates multiple grasping poses for deformable objects on a large-scale synthetic dataset. Qualitative sim-to-real experiments for such dataset, show that their simulator captures the general deformation behavior of objects during grasping. Similarly, Lu et al. (2024) introduces a simulation environment and benchmark for deformable object and garment manipulation, incorporating static scans of real-world objects to generate simulation models. Notably, they also scan 3 plush toys in static configurations. However, careful system identification and parameter tuning are necessary to achieve higher sim-to-real fidelity for synthetic datasets.

On the other hand, real-world data collection opens up the door to better capture the complex behavior of deformable objects. Current real-world datasets focus mostly on RGB images. HMDO (Xie et al., 2023) also provides real-world 3D meshes for objects undergoing deformation due to hand manipulation. However, they fell short of providing point cloud or force contact information. Chen et al. (2022) provides points clouds and force contact information but it does not perform 3D mesh reconstruction and the deformations are only globally produced using an airstream. Li et al. (2024) offer a large number of frames, however, the object diversity in their dataset is limited. Zhang et al. (2024) presents a pilot dataset with only one type of deformable object under quasi-static deformation, limited camera views, and no reported interaction forces.

In a departure from other datasets, PokeFlex offers a more comprehensive list of features including; 3D meshes, point clouds, contact forces, higher diversity of objects, and multiple types of deformations as detailed in Table 2. For simplicity, we report only the effective number of paired time frames in our table, in contrast to what is reported by Xie et al. (2023) and Li et al. (2024), where the total number of samples is computed as the number of time frames times the number of cameras.

**Data-driven mesh reconstruction methods** vary widely in terms of the input data modalities they employ. Previous approaches that rely on point clouds to predict deformations are typically trained on synthetic data (Amin Mansour et al., 2024; Lei & Daniilidis, 2022; Niemeyer et al., 2019). While synthetic training data offers controlled and dense point cloud representations, it often leads to a sim-to-real gap as real-world point cloud measurements tend to be noisy and sparse, especially in dynamic and unstructured environments. In contrast, methods using single images as input have gained attention for their real-world reconstruction capability without the need for depth information (Wang et al., 2021; Jack et al., 2019; Kanazawa et al., 2018). However, many of these image-based approaches are not optimized for online inference, making them unsuitable for downstream applications in robotics, where online feedback is essential. For instance, Xu et al. (2024) proposes an instant image-to-3D framework to generate high-quality 3D assets, but requires up to 10 seconds per frame, limiting its practicality for scenarios demanding real-time processing.

## 3 METHODOLOGY

### 3.1 DATA ACQUISITION

The PokeFlex dataset involves the acquisition of deformations under two different protocols (i) poking and (ii) dropping. For the poking protocol, we use a robotic manipulator that pokes objects with

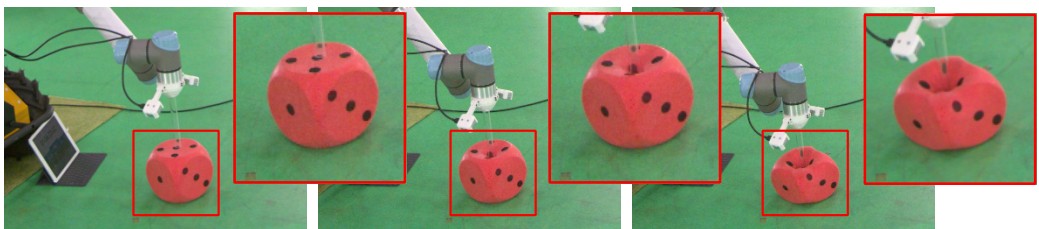

Figure 3: Sample frames from a poking sequence, with a close-up onto the foam dice.

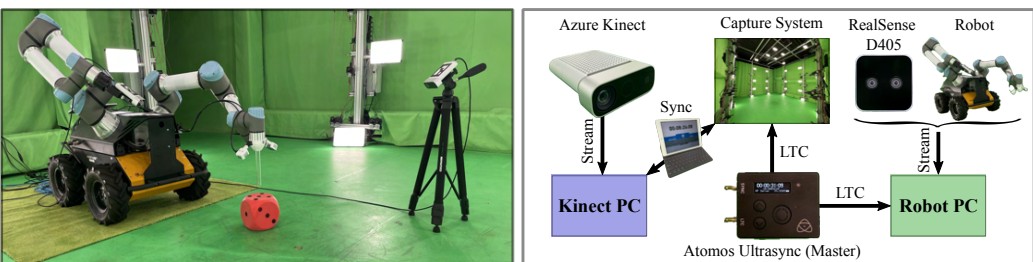

Figure 4: **Left**: Robotic manipulator positioned inside MVS with external lower-cost camera sensors during a poking sequence. **Right**: Overview of the system architecture to capture PokeFlex data.

a transparent acrylic stick multiple times along a randomly oriented horizontal vector (Figure 3). The dataset also provides the CAD model for the mounting tool, which holds two RealSense cameras and a 192 mm long acrylic stick with a radius of 10 mm. For the dropping protocol, objects are attached to a light nylon cord at approximately 2 m height and captured in a free-fall drop onto a flat surface. We record data at 30 fps and 60 fps for the poking and dropping protocols, respectively. We leverage a professional multi-view volumetric capture system (MVS), consisting of 106 cameras (53 RGB / 53 infrared) with 12 MP resolution.

For the poking protocol, we integrated and synchronized additional hardware to the MVS capture system to ensure temporally aligned data capture across all modalities. The additional hardware includes the robot manipulator and four additional RGB-D cameras: two Azure Kinect cameras to capture the scene from opposing viewpoints, and two Intel RealSense D405 cameras mounted on the robot's end-effector. The robot logs end-effector poses, interaction forces and torques at 120 Hz, while these four cameras record RGB-D data at 30 Hz.

To synchronize devices, we rely on a Linear Timecode (LTC) signal provided by an Atomos Ultrasync device. The cameras of the MVS have a leader/follower architecture, where the internal clocks of the follower cameras are synchronized to one single leader camera, which reads the LTC signal. In addition to the MVS control system, we use two desktop PCs to read the additional data streams: a Robot PC that reads the robot data and the streams of the two RealSense D405 cameras and a dedicated Kinect PC that reads the streams of the two Azure Kinect devices. The robot PC is synchronized with the capture system by reading the same LTC signal provided by the Atomos Ultrasync device. The Kinect cameras are hardware-synchronized with each other. Their synchronization with the capture system is achieved retrospectively by comparing the current time-code displayed on a screen in the camera frames of the Kinect and the camera frames of the capture system. An overview of the architecture is illustrated on Figure 4 (Right).

We utilize a system similar to that described by Collet et al. (2015) to reconstruct the meshes and textures of the objects under deformation. When recording at 30 fps, the MVS generates approximately 27 GB of raw data per second. This data is then processed using commercial software provided by Acturus Studio on 10x On-Prem Nodes servers, achieving an output rate of approximately one 3D frame per minute. The authors curate the reconstructed meshes and textures to ensure that only the deformable objects are retained in the scene.

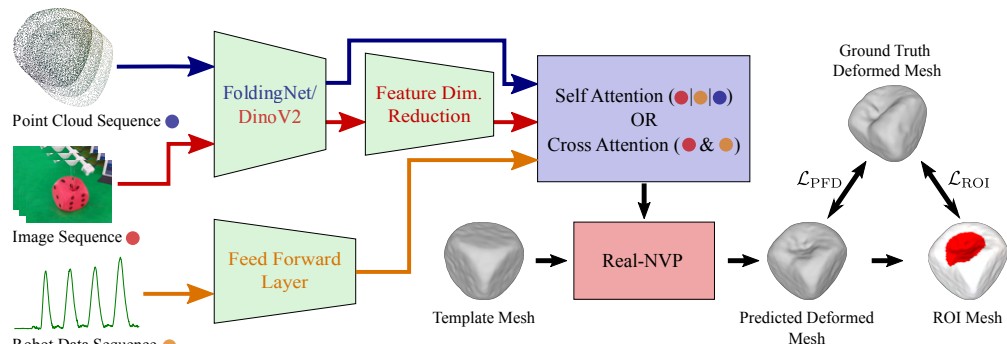

Figure 5: Superimposed representation of the proposed network architectures for ingesting the multi-modal PokeFlex data to predict deformed mesh reconstruction.

## 3.2 LEARNING-BASED MESH RECONSTRUCTION

We leverage PokeFlex to train models for template-based mesh reconstruction, where we infer the deformation of the rest-state mesh of an object using various combinations of input data modalities: sequences of images, point clouds, and/or robot data. Figure 5 illustrates the building blocks that we used to generate different architectures depending on the input modalities.

At a high level, we use three main common components for all models: an encoder for extracting features from an input modality, an attention mechanism for exploiting temporal information from the sequences, and a conditional Real-NVP (Amin Mansour et al., 2024) for predicting the offsets of template vertices, yielding the predicted deformed mesh. Real-NVP utilizes a series of conditional coupling blocks, each defined as a continuous bijective function. This continuous bijective operation ensures that the model is homeomorphic, which allows stable deformation of a template mesh while preserving its topology.

**Image input:** For pipelines using images as input, we use a DinoV2 vision transformer to extract embeddings of each image frame. In particular, we use a DinoV2-small model, pretrained via distillation from the largest DinoV2 transformer presented in Oquab et al. (2023) (LVD-142M dataset). The embedding dimension is later reduced using a 1D convolutional layer and a subsequent fully connected layer (Feature Dim. Reduction block in Figure 5).

**Point cloud input:** When using point clouds, we leverage a FoldingNet encoder (Yang et al., 2018) for representation learning, which is trained end-to-end together with the attention mechanism and the conditional-NVP.

**Robot data input:** To fuse the robot data, we concatenate the measured end-effector forces and the position of the interaction point. The concatenated data is later fed into a single fully connected layer, to match the dimensionality of the embeddings used for the attention mechanisms.

A self-attention mechanism is employed for variations of the architecture in Figure 5 that use a single data modality as input. In contrast, a cross-attention mechanism is applied when handling multiple data modalities simultaneously. For the experiments presented in the results section, we use cross-attention to handle a mixture of image sequences and robot data sequences as input. However, other combinations of input data are also possible.

All architectures are end-to-end trained using the same loss. We include the weights of the DinoV2 transformer during backpropagation for finetuning. The main point face distance (PFD) criterion $\mathcal{L}_{\text{PFD}}$ accounts for the global deformation of the objects, which computes the average squared distance $d(\boldsymbol{p}, \boldsymbol{f})$ from the set of sampled points $\boldsymbol{p}_i \in \mathcal{P}$ on the predicted mesh to the nearest faces in the set of triangular faces $\boldsymbol{f}_i \in \mathcal{F}$ of the ground truth mesh and vice versa (eq. (1)). Moreover, to deal with the local deformations generated in the poking region, we add a region-of-interest (ROI) loss $\mathcal{L}_{\text{ROI}}$ (eq. (2)) that computes the unidirectional chamfer distance from the points $\boldsymbol{p}_i$ in the ROI to the set of sampled points $\boldsymbol{q}_i \in \mathcal{Q}$ of the ground truth mesh. The ROI is defined using the indicator function $\mathbb{I}(\mathcal{C}(\boldsymbol{p}_i))$, which evaluates to 1 if point $\boldsymbol{p}_i$ is close enough to the contact point $\boldsymbol{t}$ according

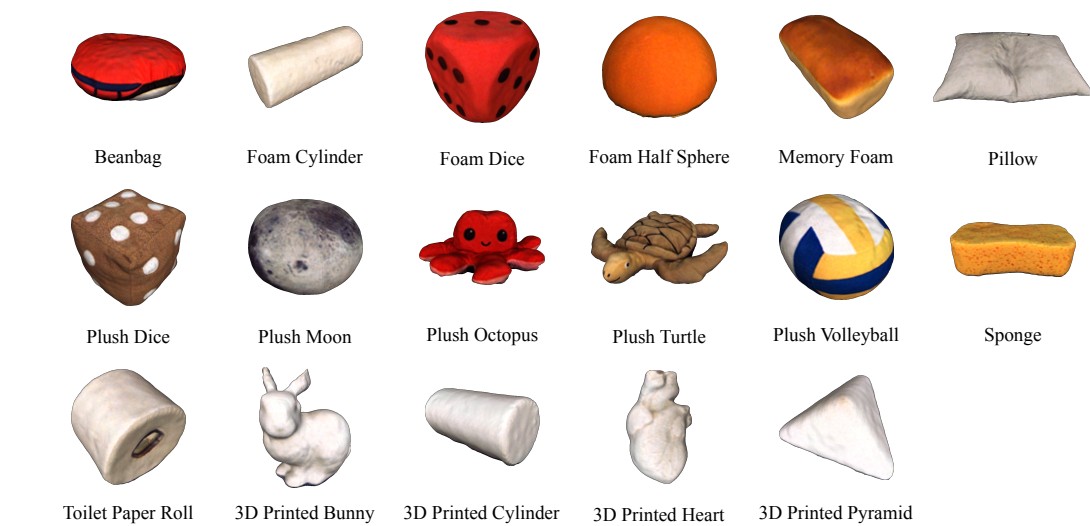

Beanbag Foam Cylinder Foam Dice Foam Half Sphere Memory Foam Pillow

Plush Dice Plush Moon Plush Octopus Plush Turtle Plush Volleyball Sponge

Toilet Paper Roll 3D Printed Bunny 3D Printed Cylinder 3D Printed Heart 3D Printed Pyramid

Figure 6: Rest-state reconstructed 3D meshes of all 17 objects featured in the PokeFlex dataset.

to a threshold $\epsilon$, and if the minimum vertical component of the contact point $\boldsymbol{p}_{i,y}$ is bigger than the minimum vertical coordinate across all the vertices $y_{\min}$ scaled by a factor (eq. (3)).

$$\mathcal{L}_{\text{PFD}} = \frac{1}{|\mathcal{P}|} \sum_{\boldsymbol{p}_i \in \mathcal{P}} \min_{\boldsymbol{f}_j \in \mathcal{F}} d(\boldsymbol{p}_i, \boldsymbol{f}_j) + \frac{1}{|\mathcal{F}|} \sum_{\boldsymbol{f}_j \in \mathcal{F}} \min_{\boldsymbol{p}_i \in \mathcal{P}} d(\boldsymbol{f}_j, \boldsymbol{p}_i) \,, \tag{1}$$

$$\mathcal{L}_{\text{ROI}} = \frac{1}{|\mathcal{P}|} \sum_{\boldsymbol{p}_i \in \mathcal{P}} \mathbb{I}(\mathcal{C}(\boldsymbol{p}_i)) \cdot \min_{\boldsymbol{q}_j \in \mathcal{Q}} \|\boldsymbol{p}_i - \boldsymbol{q}_j\|^2 \,, \tag{2}$$

$$\mathcal{C}(\boldsymbol{p}_i) = (\|\boldsymbol{p}_i - \boldsymbol{t}\| \leq \epsilon) \wedge (\boldsymbol{p}_{i,y} > 0.2 \cdot y_{\min}) \,. \tag{3}$$

The total loss is then set as $\mathcal{L} = \mathcal{L}_{\text{PFD}} + 0.5\,\mathcal{L}_{\text{ROI}}$.

## 4 RESULTS

### 4.1 DATASET

The PokeFlex dataset comprises 17 deformable objects (Figure 6), including 13 everyday items as well as 4 objects that are 3D printed with a soft thermoplastic polyurethane filament. Even though the everyday objects in our dataset can be purchased from global vendors, their availability is not guaranteed worldwide. Therefore, to enhance the usability of our dataset we include deformable 3D printed objects, providing print files and detailed specifications for reproducibility. The 3D printed objects include the Stanford bunny (Turk, 1994), a cylinder, a heart (Noor et al., 2019), and a pyramid. Further details about the 3D printing can be found in Appendix A.1.

The dimensions and the weights of the PokeFlex objects range from 7 cm to 58 cm and from 22 g to 1 kg, respectively. Furthermore, using Hooke's law and applying RANSAC for linear regression to avoid outliers, we estimated the objects' stiffnesses to be in the range of 148–3,879 N/m.

For the poking protocol, we recorded 4-8 sequences with a duration of 5-6 seconds at 30 fps for each object. Similarly, for the dropping protocol, we recorded 3 sequences of 1 second at 60 fps for each object. Figure 7 shows two reconstructed sequences for poking and dropping. In the case of the poking sequences, each frame includes synchronized and paired data from all modalities, as illustrated in Figure 2.

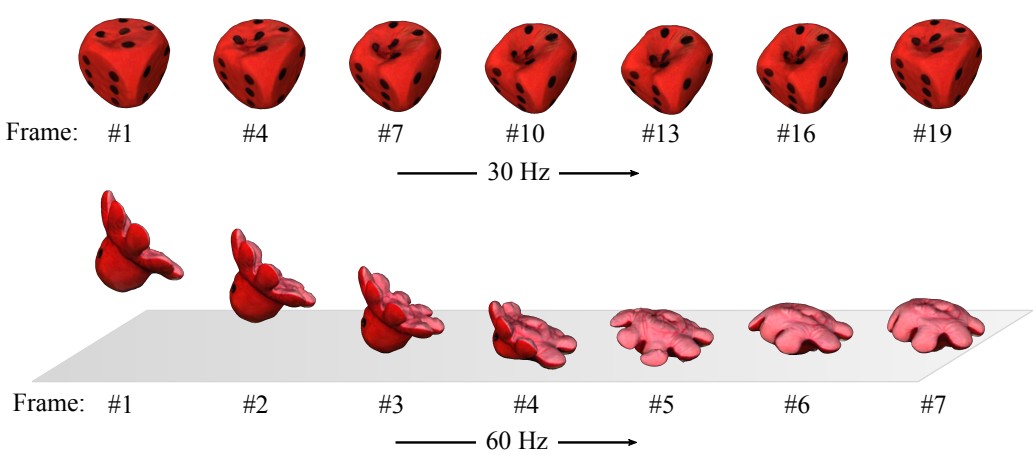

Figure 7: **Top:** Mesh reconstructions of foam dice for a poking sequence shown in every third frame. **Bottom:** Mesh reconstructions of plush octopus for a dropping sequence.

The total number of reconstructed frames used to generate ground-truth data was 19k, which comprises 16.1k frames for the poking sequences and 3.1k frames for the dropping sequences. Considering the different modalities, the total of PokeFlex amounts to more than 240k samples. It is worth noting that after curating the frames of the poking sequences, i.e., discarding the frames where the robot arm is not in contact with the objects, the total number of active paired poking frames sum up to 8.1k. A summary of the physical properties of the objects, as well as a per-object list of the recorded frames under deformation for the poking sequences, is presented in Appendix A.2. For the dropping protocol, we recorded 180 frames per object.

## 4.2 EVALUATION OF LEARNING-BASED RECONSTRUCTION

**Overview of training data.** In the following experiments, we exclusively used poking sequences from the dataset because of the higher diversity of input data modalities available. The input sequence length was set to 5, chosen heuristically for better performance. The train-validation split was generated by randomly choosing one recording sequence per object as the validation set.

**Metrics.** During training, we reposition and re-scale all meshes into a cube of unit size ($[-0.5, 0.5]^3$) to maintain a consistent scale across all objects. The losses $\mathcal{L}_{\mathrm{PFD}}$ and $\mathcal{L}_{\mathrm{ROI}}$ are computed in this normalized scale. Additionally, we calculate the relative point-to-face distance (RPFD) by dividing $\mathcal{L}_{\mathrm{PFD}}$ by the average point-to-face distance between the template mesh $M_{\mathrm{T}}$ and the ground truth mesh $M_{\mathrm{GT}}$. An RPFD value below 1 indicates that the predicted deformed mesh $M_{\mathrm{P}}$ is closer to the ground truth than the undeformed template, with smaller values indicating better accuracy.

To further assess the prediction accuracy, we evaluate two additional metrics between the predicted mesh and the ground truth mesh in their original scale: the unidirectional L1 Norm Chamfer Distance $\mathrm{CD}_{\mathrm{UL1}}$ (eq. (4)) and the volumetric Jaccard Index $J$ (eq. (5)), which we defined in terms of the volume operator $V$. The two metrics provide insights into the L1 Norm surface distance and the volume overlap ratio, respectively.

$$\mathrm{CD}_{\mathrm{UL1}} = \frac{1}{|\mathcal{P}|} \sum_{\boldsymbol{p}_i \in \mathcal{P}} \min_{\boldsymbol{q}_j \in \mathcal{Q}} \|\boldsymbol{p}_i - \boldsymbol{q}_j\|_1 \,, \tag{4}$$

$$J(\boldsymbol{M}_{\mathrm{A}}, \boldsymbol{M}_{\mathrm{B}}) = \frac{V(\boldsymbol{M}_{\mathrm{A}} \cap \boldsymbol{M}_{\mathrm{B}})}{V(\boldsymbol{M}_{\mathrm{A}} \cup \boldsymbol{M}_{\mathrm{B}})} \,. \tag{5}$$

**Learning from RGB images of different cameras.** In this experiment, we train different models to predict meshes using sequences of RGB images only. Each model was trained for a specific camera, namely Volucam (capture system), Intel RealSense D405, Azure Kinect, and a Virtual camera

Table 3: Mean prediction performance for all models trained on a single viewpoint from different cameras for all objects. Arrows indicate that a better performance is either higher ↑ or lower ↓.

| Input | $\mathcal{L}_{\textbf{PFD}} \cdot 10^3 \downarrow$ | $\mathcal{L}_{\textbf{ROI}} \cdot 10^3 \downarrow$ | $\textbf{RPFD} \downarrow$ | $\textbf{CD}_{\textbf{UL1}}\textbf{[mm]} \downarrow$ | $J(M_{\textbf{P}}, M_{\textbf{GT}}) \uparrow$ |
|---|---|---|---|---|---|
| Volucam | **6.69** | 8.38 | 0.698 | **7.433** | 0.799 |
| RealSense | 7.91 | **7.37** | **0.693** | 7.675 | **0.806** |
| Kinect | 11.20 | 9.79 | 0.839 | 8.505 | 0.767 |
| Rendered | 12.97 | 12.28 | 0.826 | 8.761 | 0.761 |
| Brighter RealSense | 14.57 | 13.25 | 0.853 | 9.126 | 0.754 |
| Darker RealSense | 13.89 | 11.74 | 0.957 | 9.831 | 0.736 |

(Images rendered from ground truth mesh). The viewpoint of each camera is different. Additionally, we provide experiments evaluating the robustness of our model for varying lighting conditions. This was achieved by adjusting each channel, by a constant value of ±20, on a scale of 0-255, and introducing noise drawn from a normal distribution with a standard deviation of 10. For training, we use all objects. The performance of the different models is reported in Table 3. The training hyperparameters used for this and the following experiments are reported in Appendix A.3.

**Learning from different data modalities.**

In this experiment, we train different mesh prediction models from sequences of different input modalities. Same as in the previous experiment, we trained multi-object models using all 17 objects from the dataset. Detailed performance for the evaluated data modalities can be found in Table 4. Inference rates across different data modalities, detailed in Appendix A.4, range from 106 Hz to 215 Hz for dense point clouds and forces, respectively. Figure 8 shows examples of predicted meshes with different levels of reconstruction quality obtained using a multi-object model trained from image-sequences only. Additionally, Appendix A.5 reports a detailed breakdown of the per-object performance for models trained from sequences of images, images + robot data, and point clouds.

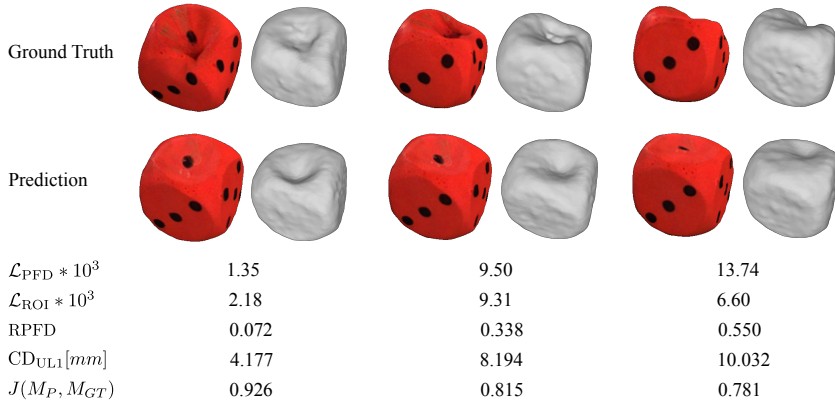

| | | | |
|---|---|---|---|
| $\mathcal{L}_{\text{PFD}} * 10^3$ | 1.35 | 9.50 | 13.74 |
| $\mathcal{L}_{\text{ROI}} * 10^3$ | 2.18 | 9.31 | 6.60 |
| RPFD | 0.072 | 0.338 | 0.550 |
| $\text{CD}_{\text{UL1}}[mm]$ | 4.177 | 8.194 | 10.032 |
| $J(M_P, M_{GT})$ | 0.926 | 0.815 | 0.781 |

Figure 8: Examples of deformation predictions for a foam dice and their corresponding metrics. Meshes are rendered side by side with and without texture to highlight the deformation in the ROI.

## 5   DISCUSSION

**Quality of ground-truth meshes.** The overall geometry of the objects in the dataset, in static configurations, is well captured by the meshes reconstructed with the MVS as shown in Figure 6, even though the system's intended use is the reconstruction of human-size objects. Furthermore, the proposed poking protocol, using a transparent acrylic stick, helps prevent occlusions at the contact point, leading to detailed reconstruction of objects even when they undergo deformations, as can be seen in Figure 9 (Left). However, reconstruction of fine-grained details for smaller objects such as the 3D-printed Stanford armadillo (Curless & Levoy, 1996) remains challenging with the current setup of the professional capture system, as seen in Figure 9 (Right). Better fine-grained reconstruction results can be expected by rearranging the cameras in a smaller workspace.

Table 4: Mean prediction performance for proposed model configurations trained on all objects.

| Input | $\mathcal{L}_{\textbf{PFD}} \cdot 10^3 \downarrow$ | $\mathcal{L}_{\textbf{ROI}} \cdot 10^3 \downarrow$ | **RPFD** $\downarrow$ | $\textbf{CD}_{\textbf{UL1}}$[mm] $\downarrow$ | $J(M_{\textbf{P}}, M_{\textbf{GT}}) \uparrow$ |
|---|---|---|---|---|---|
| Images | 6.69 | 8.38 | 0.698 | 7.433 | 0.799 |
| Robot data | 7.43 | 5.80 | 0.847 | 8.014 | 0.785 |
| Images + robot data | 5.39 | 5.10 | 0.594 | 6.642 | 0.821 |
| Dense synthetic point clouds (5k points) | **4.76** | **4.92** | 0.569 | **6.338** | **0.831** |
| Sparse synthetic point clouds (100 points) | 6.14 | 5.61 | 0.577 | 6.569 | 0.815 |
| Kinect point clouds | 6.17 | 6.56 | 0.592 | 6.619 | 0.807 |
| Kinect point clouds + robot data | 6.86 | 6.18 | **0.539** | 6.613 | 0.817 |

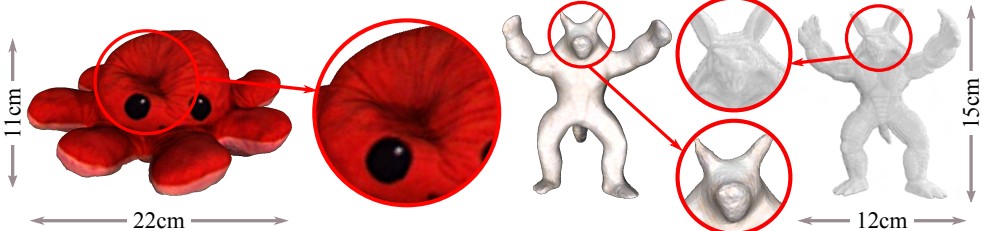

Figure 9: Examples of reconstructed ground truth meshes for medium (**Left**) and small (**Right**) size objects in deformed states. Reconstruction of fine-grained details is a limitation of our current setup (close-up views on the Right).

**Estimated stiffness.** The estimated stiffness that we provide for the featured objects is only intended to offer insights into the range of material properties included in PokeFlex. The simple linear interpolation method using RANSAC can successfully characterize the linear Hookean behavior of objects such as the foam or plush dice shown in Figure 10. More sophisticated approaches, like the ones presented by Sundaresan et al. (2022) and Heiden et al. (2021) leveraging differentiable simulation, are needed to better characterize the nonlinear behavior exhibited by thinner objects such as the plush turtle.

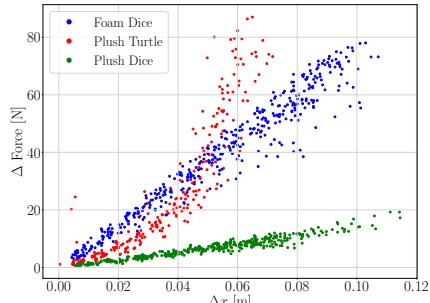

Figure 10: Acting force vs. end effector displacement, shown across all frames for three objects from PokeFlex.

**Learning from RGB images of different cameras.** The results in Table 3 show that the best performance is obtained using RGB images coming from the Volucam or the RealSense cameras. The model trained from the Volucam cameras performs the best in terms of the validation loss $\mathcal{L}_{\textbf{PFD}}$ and the chamfer distance $\textbf{CD}_{\textbf{UL1}}$. The model trained from RealSense images performs best with respect to all other losses and metrics ($\mathcal{L}_{\textbf{ROI}}$, RPFD, $\text{CD}_{\text{UL1}}$, $J(M_{\textbf{P}}, M_{\textbf{GT}})$). In particular, the high performance of the latter model in terms of $\mathcal{L}_{\textbf{ROI}}$ can be attributed to the proximity of the RealSense camera relative to the ROI. Furthermore, regardless of the variability in terms of the validation losses for different cameras, the performance measured by the chamfer loss remains within a few millimeters of the best-performing model, showing that good-performing models can be trained using camera sensors that are external to the professional capture system, even if they have different viewpoints.

**Multi-object mesh reconstruction from different modalities.** Table 4 shows that the dense synthetic point clouds yield the best performance among all data modalities. A drop in performance is observed for the sparser synthetic point clouds, and the noisier point clouds captured by the Kinect. The model trained from images and robot data achieves the second-best performance overall, outperforming the model trained from images only, showcasing the importance of the robot data, and hinting at the effectiveness of our cross-attention mechanism. Combining robot data with the Kinect

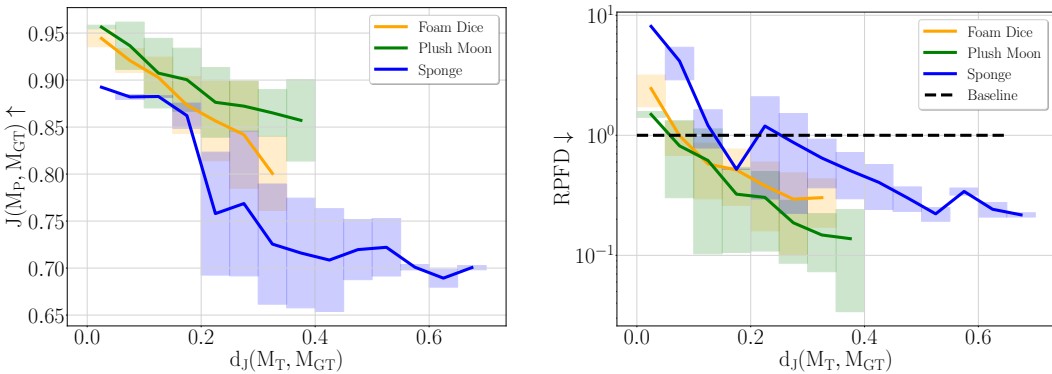

Figure 11: Validation accuracy for image-based mesh reconstruction, evaluated by Jaccard Index $J$ (**Left**) and RPFD (**Right**), plotted against the deformation level quantified by Jaccard distance $d_J$.

point clouds also leads to performance improvements relative to only using the Kinect point clouds, however the performance gains are more subtle.

To analyze the levels of accuracy across multiple objects, we focus on the image-based mesh reconstruction model. Figure 11 shows $J(M_P, M_{GT})$ and RPFD for only 3 objects separately, for clarity of visualization. The horizontal axis is the Jaccard distance, which indicates the level of deformation of the ground truth mesh with respect to the rest-state template mesh, defined as $d_J(M_T, M_{GT}) = 1 - J(M_T, M_{GT})$. Figure 11 shows that the best prediction performance is obtained for the plush moon, having the highest Jaccard Index and the lowest RPFD. The corresponding results for all objects are reported in Appendix A.5, together with the histograms that show the samples distribution.

In contrast, for low deformation regimes (small values of $d_J(M_T, M_{GT})$), the foam cylinder exhibits a lower accuracy, reaching values higher than 1 for the RPDF metric. Such high values correspond to a performance worse than that of predicting the rest-state mesh. Both performance metrics reported in Figure 11 show, overall, a negative correlation with the Jaccard distance for all objects, indicating that the prediction accuracy of our models decreases for larger deformations. Further experiments, testing the generalization of 3D mesh reconstruction to unseen objects are reported in Appendix A.7.

## 6 CONCLUSION

This paper introduced PokeFlex, a new dataset that captures the behavior of 17 deformable volumetric objects during poking and dropping. The focus is on volumetric objects, while thin clothing items or thin cables are not considered in the dataset. Compared to previously existing datasets, we provide a wider range of paired and annotated data modalities, which are supplemented with data streams from lower-cost camera sensors. In an effort to enhance reproducibility, the objects included in our dataset can be either purchased from global providers or 3D printed with our open-source models. The 3D printed objects also allow for finer control over their expected behavior through knowledge of their material properties and internal structures, especially useful for sim-to-real transfer.

Using different combinations of the data modalities provided in PokeFlex, we train and benchmark a list of baseline models for the task of multi-object template-based mesh reconstruction. In doing so, we present a list of suitable criteria for evaluating PokeFlex.

We are excited about the potential of PokeFlex to inspire new research directions in deformable object manipulation and to serve as a foundational resource for the robotics community. With its rich, multimodal data and its focus on reproducibility, we believe that PokeFlex will drive innovation in both simulation-based and real-world applications of deformable object manipulation. This includes better material parameter identification to fine-tune simulators, viewpoint-agnostic online 3D mesh reconstruction methods, and policy learning for manipulation tasks. As we continue to expand the dataset and explore new possibilities, we anticipate that PokeFlex will become an invaluable tool for researchers developing next-generation techniques. We look forward to sharing this dataset with the community and fostering collaborations that push the boundaries of robotics research.

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

# A APPENDIX

## A.1 3D PRINTING DETAILS

All 3D printed objects were printed using thermoplastic polyurethane (TPU) Filaflex Shore 60A Pro White filament on Prusa MK3S+ and Prusa XL 3D printers equipped with 0.4mm nozzles. The mechanical properties of the filament are presented in Table 5.

Table 5: Mechanical Properties of Filaflex shore 60A Pro TPU provided by the manufacturer.

| Mechanical properties | Value | Unit | Test method according to |
|---|---|---|---|
| Tensile strength | 26 | MPa | DIN 53504-S2 |
| Stress at 20% elongation | 1 | MPa | DIN 53504-S2 |

The printing parameters of the 3D printed objects are summarized in Table 6, where the infill used for all objects is the isotropic gyroid pattern with uniform properties and behavior in all directions. Example of the gyroid pattern can be seen in Figure 12.

Table 6: Printing parameters of 3D printed objects featured in the PokeFlex dataset.

| Object | Infill density [%] | Layer thickness [mm] | Perimeters | Bottom layers | Top layers |
|---|---|---|---|---|---|
| Bunny (Turk, 1994) | 10 | 0.2 | 3 | 3 | 3 |
| Cylinder | 10 | 0.15 | 2 | 3 | 3 |
| Heart (Noor et al., 2019) | 10 | 0.2 | 3 | 3 | 3 |
| Pyramid | 8 | 0.2 | 3 | 3 | 3 |

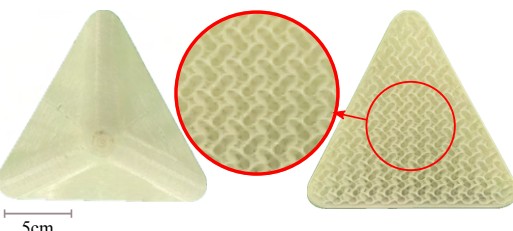

5cm

Figure 12: Top (**Left**) and bottom (**Right**) view of 3D printed pyramid, with a close-up view of the interior gyroid infill pattern.

## A.2 PROPERTIES OF FEATURED OBJECTS

In Table 7, we summarize the physical properties and the number of frames per object. The Frames column of the table presents the total captured frames of the poking sequences for each object, and the Deformations column gives the number of active poking frames after the data curation, i.e., discarding the frames where the robot arm is not in contact with the objects. It is worth noting that we report only the effective number of paired time frames in our table, in contrast to the total number of samples, which is computed as the number of time frames multiplied by the number of cameras.

Table 7: Physical properties of objects featured in the PokeFlex dataset. Dimensions of sphere-like objects are described by their diameter (D). Cylinder-like objects are characterized by their diameter (D) and height (H). For objects with irregular or complex shapes, dimensions are provided using a bounding box defined by length (L), width (W), and height (H). Stiffness of the objects is estimated according to the method described in Section 4.1.

| Object | Weight [g] | Dimensions [cm] | Est. stiffness [N/m] | Frames | Deformations |
|---|---|---|---|---|---|
| Beanbag | 184 | DxH: 26x9 | 523 | 1084 | 363 |
| Foam cylinder | 153 | DxH: 12x29 | 250 | 990 | 407 |
| Foam dice | 140 | L: 15.5 | 748 | 1220 | 738 |
| Foam half sphere | 41 | D: 15 | 1252 | 619 | 384 |
| Memory foam | 213 | LxWxH: 17.5x8.5x7 | 395 | 420 | 141 |
| Pillow | 975 | LxWxH: 58x50x10 | 474 | 1085 | 565 |
| Plush dice | 340 | L: 22 | 149 | 1259 | 567 |
| Plush moon | 151 | D: 17 | 366 | 959 | 517 |
| Plush octopus | 130 | LxWxH: 22x22x11 | 325 | 1085 | 525 |
| Plush turtle | 194 | LxWxH: 35x30x10 | 1035 | 930 | 427 |
| Plush volleyball | 303 | D: 22 | 323 | 1099 | 604 |
| Sponge | 28 | LxWxH: 22x12x6.1 | 1045 | 1237 | 772 |
| Toilet paper roll | 134 | DxH: 10.5x9.5 | 2156 | 600 | 295 |
| 3D printed bunny | 105 | LxWxH:13x9x15 | 950 | 1127 | 593 |
| 3D printed cylinder | 223 | DxH: 10x20 | 585 | 1020 | 574 |
| 3D printed heart | 100 | LxWxH: 16x9x10 | 1198 | 940 | 444 |
| 3D printed pyramid | 48 | LxWxH: 14.5x14.5x7 | 861 | 420 | 193 |

For the dropping protocol, we recorded 3 sequences of 1 second at 60 fps for each object, summing up to 180 time frames per object. Figure 13 shows two additional reconstructed deformed mesh sequences for dropping the foam cylinder and the pillow, respectively.

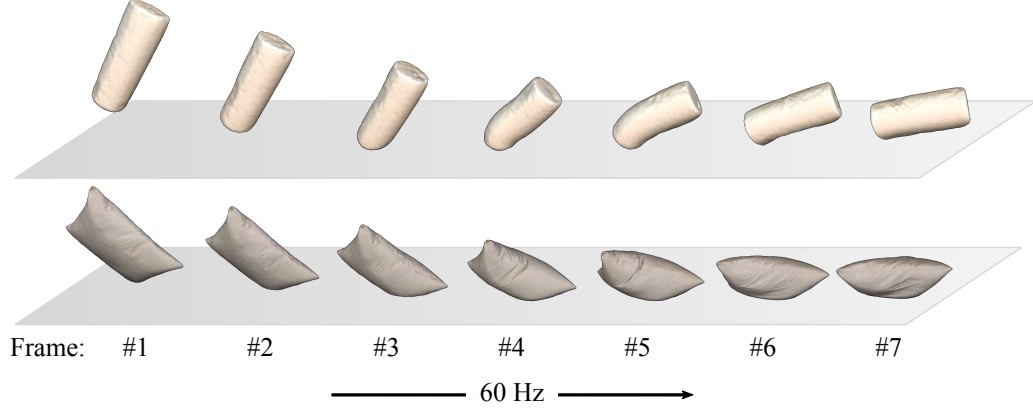

Frame:   #1      #2      #3      #4      #5      #6      #7

⟶ 60 Hz ⟶

Figure 13: Sample of mesh reconstructions of foam cylinder (**Top**) and pillow (**Bottom**) for a dropping sequence, respectively.

## A.3 TRAINING DETAILS

The hyperparameters used to train the models in Section 4.1 are listed in Table 8.

Table 8: Training hyperparameters.

| Hyperparameters | Value |
|---|---|
| Learning rate | 1e-4 |
| Batch size | 16 (5 objects) / 8 (1 object) |
| Optimizer | Adam |
| Weight decay | 5e-6 |
| Learning rate scheduler | Cosine |
| Minimum learning rate | 1e-7 |
| Epochs | 200 |

## A.4 INFERENCE SPEED FOR DIFFERENT INPUT DATA MODALITIES

Table 9 shows the measured inference rates for our five proposed models with different input data modalities. The rate is tested with an AMD Ryzen 7900 x 12 Core Processor CPU and NVIDIA GeForce RTX 4090 GPU with 24GB memory.

Table 9: Inference rate for proposed model configurations.

| Input | Inference Speed |
|---|---|
| Images | 115 Hz |
| Forces | 215 Hz |
| Images + forces | 110 Hz |
| Point clouds (5000 points) | 106 Hz |
| Point clouds (100 points) | 195 Hz |

## A.5 PER OBJECT ANALYTICS FOR LEARNING-BASED MESH RECONSTRUCTION

In the following, we present the performance metrics across all objects for several modalities. This includes models trained on images (Table 10), images + robot data (Table 11), and point clouds (Table 12). The respective values show the performance on the validation sequence for each object.

Table 10: Prediction metrics for each object for image-based mesh reconstruction. Bold values indicate better performance than the average across all objects.

| Input | $\mathcal{L}_{\mathbf{PFD}} \cdot 10^3 \downarrow$ | $\mathcal{L}_{\mathbf{ROI}} \cdot 10^3 \downarrow$ | $\mathbf{RPFD} \downarrow$ | $\mathbf{CD_{UL1}[mm]} \downarrow$ | $J(M_{\mathbf{P}}, M_{\mathbf{GT}}) \uparrow$ |
|---|---|---|---|---|---|
| Beanbag | **5.86** | 12.78 | **0.467** | 7.994 | **0.833** |
| Foam cylinder | **5.87** | **7.61** | 1.366 | 9.300 | 0.767 |
| Foam dice | **5.06** | 12.19 | 0.845 | **5.900** | **0.884** |
| Foam half sphere | **0.81** | **2.24** | **0.229** | **2.640** | **0.927** |
| Memory foam | 9.34 | 17.21 | 0.978 | **7.432** | 0.723 |
| Pillow | **1.35** | **1.88** | 0.920 | 10.70 | **0.840** |
| Plush dice | **4.80** | **5.05** | 0.823 | 9.511 | **0.869** |
| Plush moon | **3.85** | **6.73** | **0.391** | **5.437** | **0.890** |
| Plush octopus | **2.28** | **2.09** | 0.855 | **6.851** | 0.773 |
| Plush turtle | **2.10** | **0.89** | 1.174 | 9.268 | 0.732 |
| Plush volleyball | 8.96 | 15.73 | **0.290** | 9.083 | **0.835** |
| Sponge | 8.39 | **4.71** | **0.513** | 7.661 | 0.734 |
| Toilet paper roll | 24.93 | 37.33 | **0.501** | 9.055 | 0.672 |
| 3D printed bunny | 10.60 | **4.38** | **0.677** | **6.943** | 0.714 |
| 3D printed cylinder | **5.04** | **6.06** | 0.777 | **6.005** | **0.820** |
| 3D printed heart | 8.17 | **4.70** | **0.341** | **6.115** | 0.764 |
| 3D printed pyramid | **6.53** | **4.09** | 0.988 | **5.725** | 0.695 |

Table 11: Prediction metrics for each object for using the combination of images and robot data as input. Bold values indicate better performance than the average across all objects.

| Input | $\mathcal{L}_{\mathbf{PFD}} \cdot 10^3 \downarrow$ | $\mathcal{L}_{\mathbf{ROI}} \cdot 10^3 \downarrow$ | $\mathbf{RPFD} \downarrow$ | $\mathbf{CD_{UL1}[mm]} \downarrow$ | $J(M_{\mathbf{P}}, M_{\mathbf{GT}}) \uparrow$ |
|---|---|---|---|---|---|
| Beanbag | 5.70 | 7.59 | 0.810 | 8.114 | **0.825** |
| Foam cylinder | **2.10** | **1.43** | 0.623 | 6.807 | **0.844** |
| Foam dice | **1.66** | 5.96 | 0.667 | **4.276** | **0.925** |
| Foam half sphere | **0.80** | **1.12** | **0.272** | **2.622** | **0.928** |
| Memory foam | 22.27 | 23.93 | 1.585 | 10.007 | 0.588 |
| Pillow | **0.84** | **0.96** | 0.625 | 9.092 | **0.877** |
| Plush dice | **2.47** | **2.35** | 0.629 | 7.689 | **0.902** |
| Plush moon | **4.32** | **3.43** | **0.518** | **6.304** | **0.869** |
| Plush octopus | **1.23** | **1.01** | **0.536** | **5.612** | **0.823** |
| Plush turtle | **1.73** | **0.58** | 0.949 | 8.705 | 0.760 |
| Plush volleyball | **2.45** | **2.57** | **0.166** | **6.076** | **0.901** |
| Sponge | **5.29** | **1.43** | **0.340** | **6.439** | 0.791 |
| Toilet paper roll | 17.94 | 22.80 | **0.385** | 7.651 | 0.726 |
| 3D printed bunny | 8.79 | **3.86** | 0.791 | **6.284** | 0.722 |
| 3D printed cylinder | **5.21** | 5.61 | 0.666 | **5.943** | **0.822** |
| 3D printed heart | 6.79 | **4.17** | **0.294** | **5.658** | 0.789 |
| 3D printed pyramid | 5.43 | **3.61** | 0.878 | **5.617** | 0.705 |

Table 12: Prediction metrics for each object for point-cloud-based mesh reconstruction. Bold values indicate better performance than the average across all objects.

| Input | $\mathcal{L}_{\textbf{PFD}} \cdot 10^3 \downarrow$ | $\mathcal{L}_{\textbf{ROI}} \cdot 10^3 \downarrow$ | $\textbf{RPFD} \downarrow$ | $\textbf{CD}_{\textbf{UL1}}[\text{mm}] \downarrow$ | $J(M_{\textbf{P}}, M_{\textbf{GT}}) \uparrow$ |
|---|---|---|---|---|---|
| Beanbag | 9.18 | 8.63 | 1.402 | 9.608 | 0.790 |
| Foam cylinder | **0.73** | **0.67** | **0.337** | **4.701** | **0.909** |
| Foam dice | **1.98** | **2.6** | **0.414** | **4.415** | **0.924** |
| Foam half sphere | **0.32** | **1.41** | **0.148** | **2.202** | **0.952** |
| Memory foam | **4.54** | 8.26 | **0.578** | **5.008** | 0.786 |
| Pillow | **0.85** | **1.23** | 0.666 | 8.835 | **0.882** |
| Plush dice | **2.22** | **2.68** | 0.583 | 7.433 | **0.905** |
| Plush moon | **0.68** | **1.15** | **0.179** | **3.546** | **0.946** |
| Plush octopus | 4.88 | **4.68** | 1.310 | 8.937 | 0.692 |
| Plush turtle | **1.22** | **0.89** | 0.98 | 8.076 | 0.762 |
| Plush volleyball | **1.67** | **3.36** | **0.106** | **5.307** | **0.921** |
| Sponge | 5.28 | **2.55** | **0.354** | 7.237 | 0.760 |
| Toilet paper roll | 14.13 | 17.06 | **0.314** | 6.901 | 0.742 |
| 3D printed bunny | 12.22 | 7.74 | 0.883 | 7.289 | 0.665 |
| 3D printed cylinder | **3.06** | **2.36** | 0.573 | **4.834** | **0.858** |
| 3D printed heart | 9.91 | 6.73 | **0.488** | 6.475 | 0.754 |
| 3D printed pyramid | **4.29** | 6.97 | 0.653 | **4.681** | 0.765 |

## A.6 Validation accuracy for image-based mesh reconstruction.

In the plots below, we show $J(M_\mathrm{P}, M_\mathrm{GT})$ and RPFD for all objects, and the underlying distribution for each object in form of a histogram (Figure 14 - Figure 19).

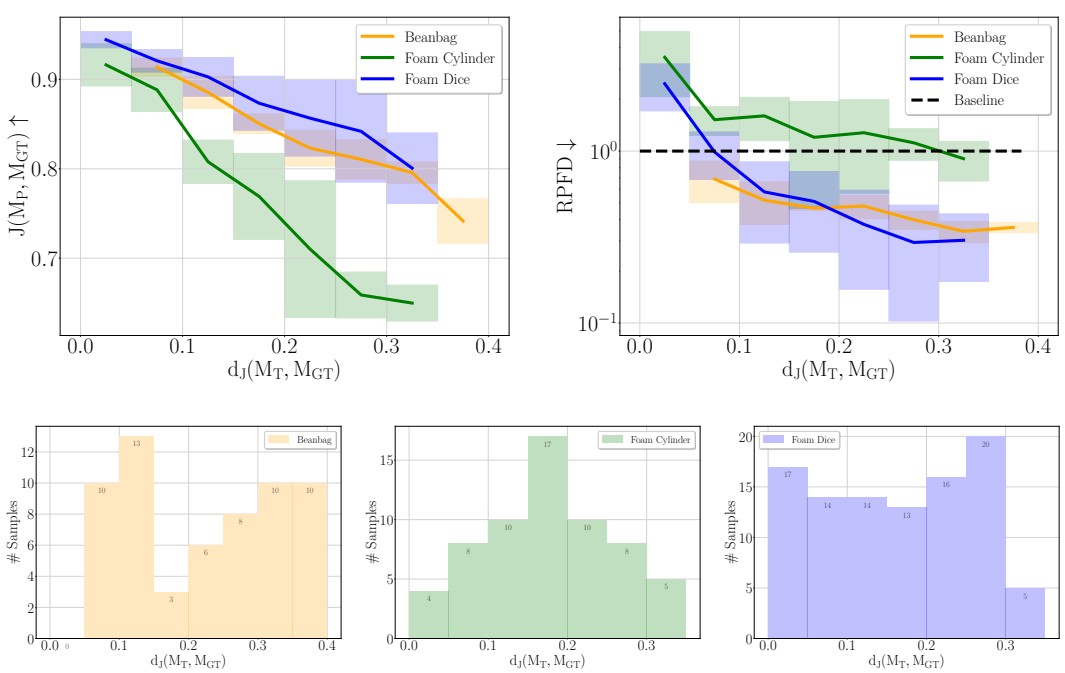

Figure 14: Beanbag, foam cylinder, foam dice.

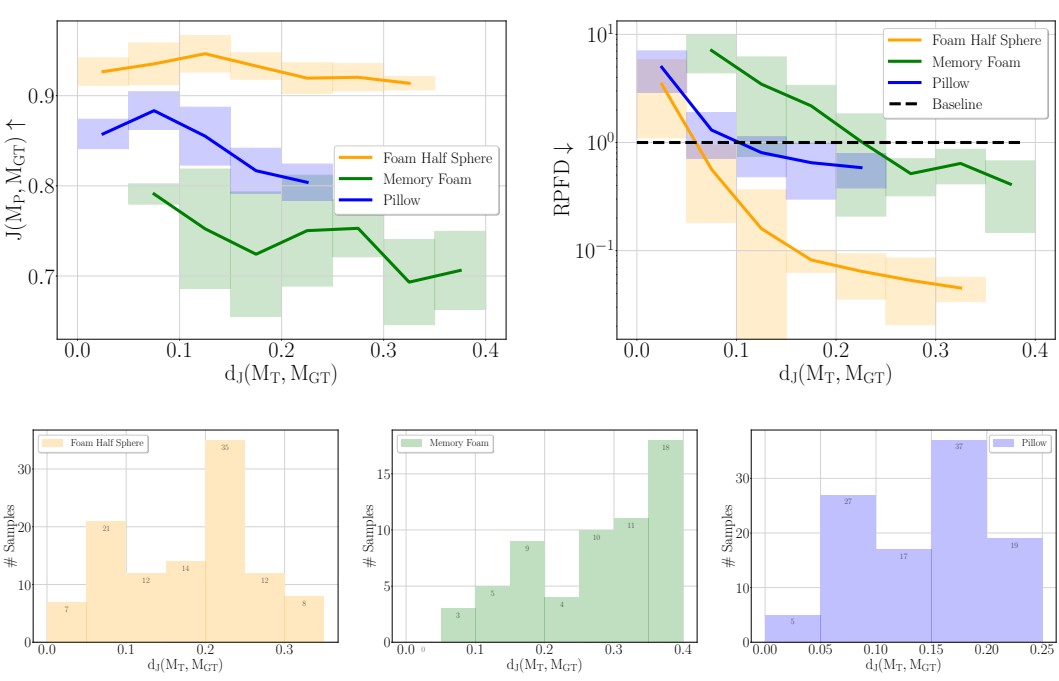

Figure 15: Foam half sphere, memory foam, pillow.

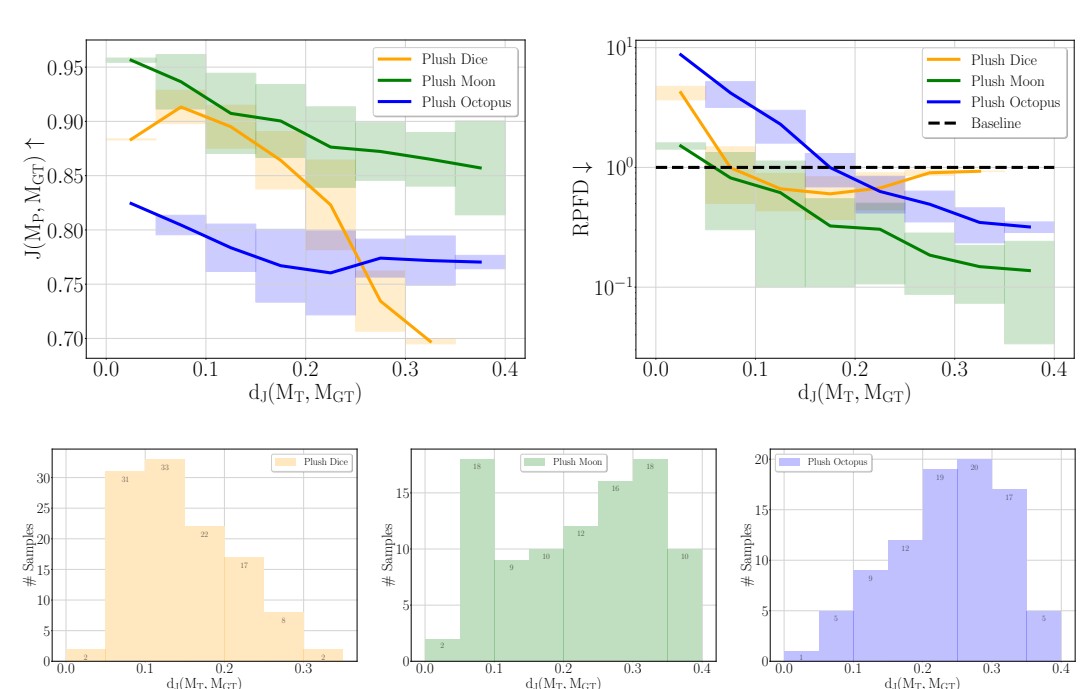

Figure 16: Plush dice, plush moon, plush octopus.

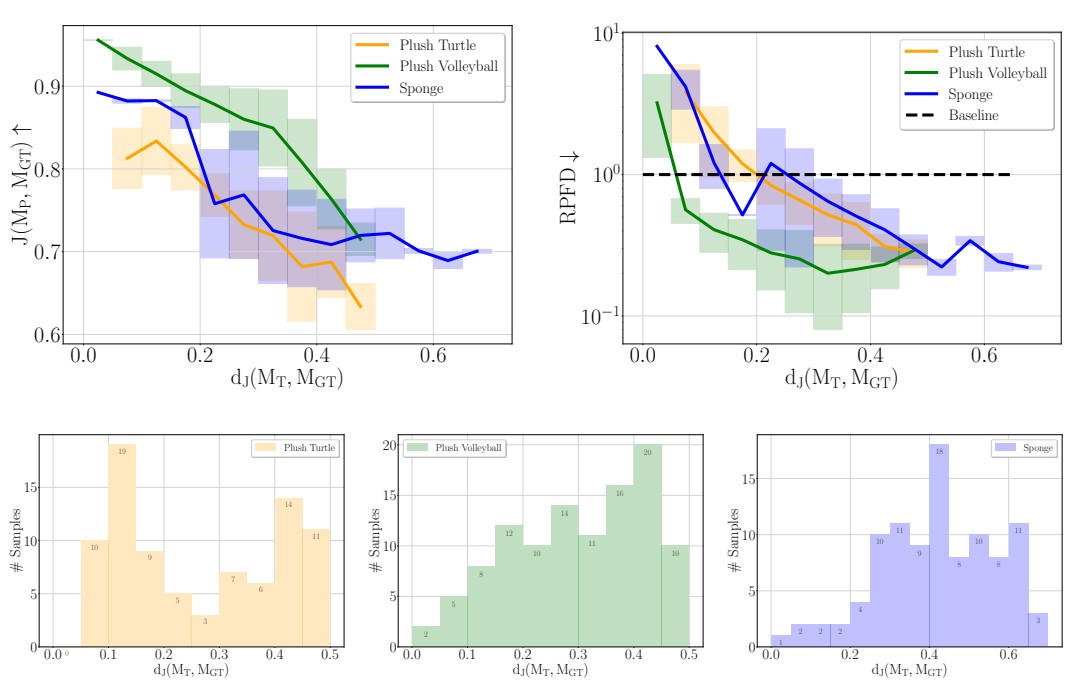

Figure 17: Plush turtle, plush volleyball, sponge.

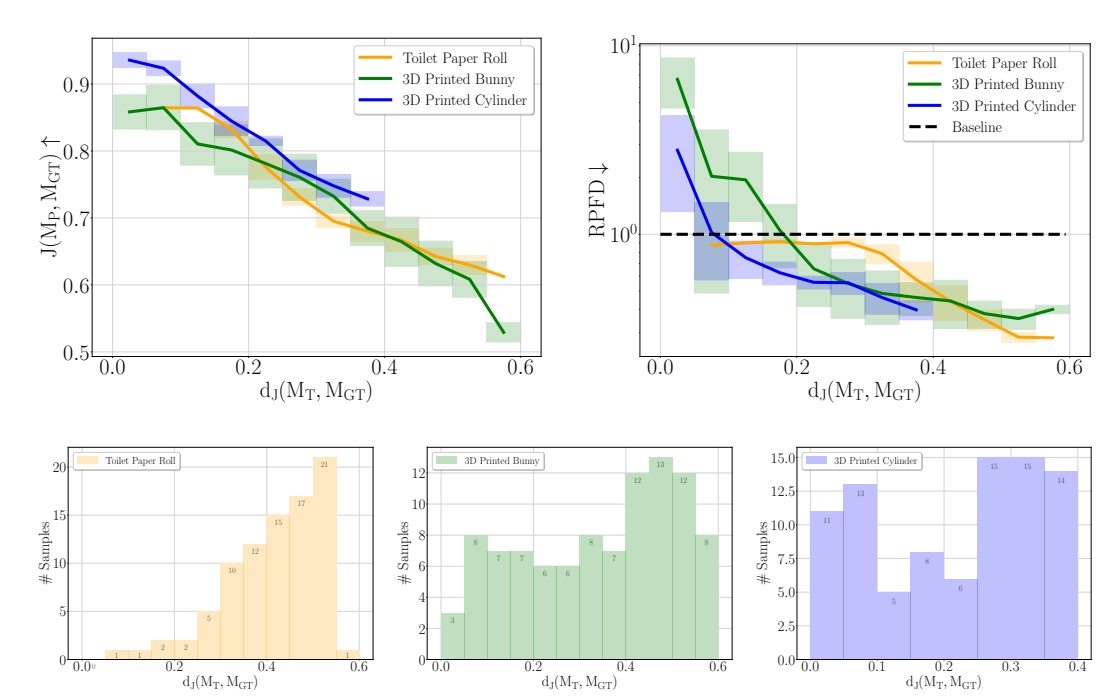

Figure 18: Toilet paper roll, 3D printed bunny, 3D printed cylinder.

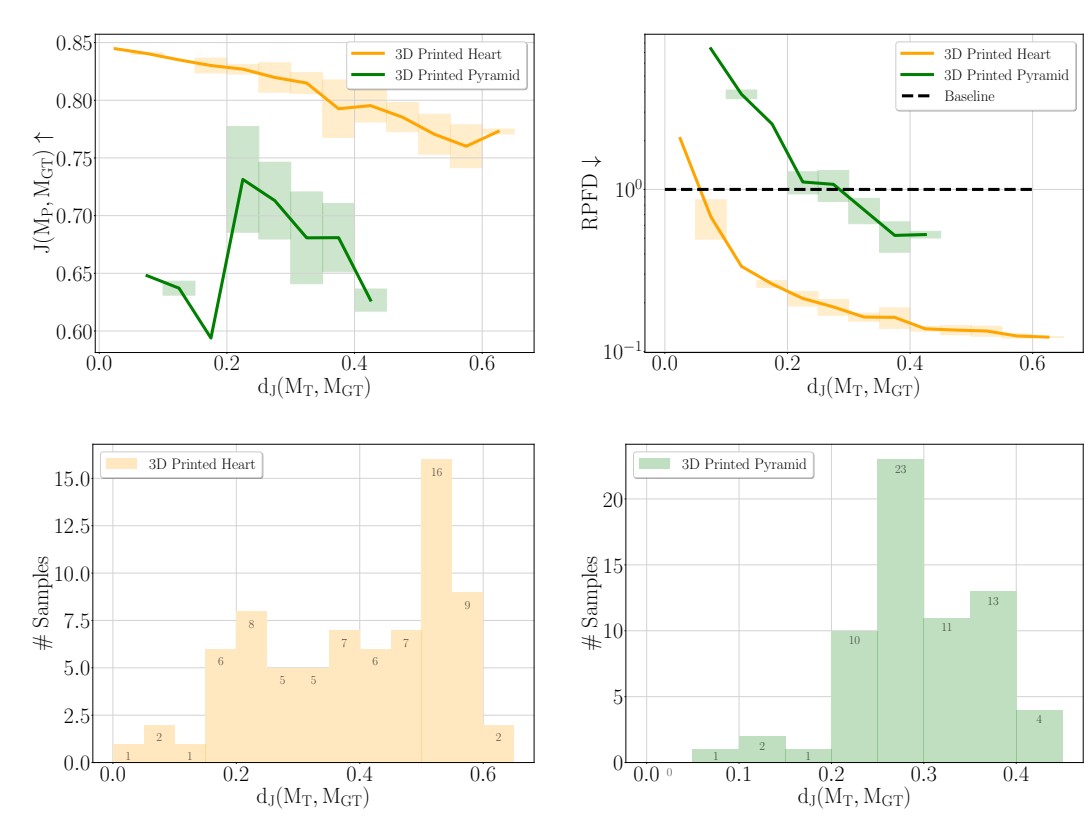

Figure 19: 3D printed heart, 3D printed pyramid.

### A.7 Generalization Performance for Learning-Based Mesh Reconstruction

In the following, we show our model's capabilities to generalize predictions. We trained models on the modalities point clouds (Table 13) and the combination of images and robot data (Table 14) on 13 different objects and evaluated on 4 unseen objects. The evaluation on unseen objects included all sequences.

Table 13: Generalization results for 4 unseen objects for point-cloud-based mesh reconstruction.

| Input | $\mathcal{L}_{\mathbf{PFD}} \cdot 10^3 \downarrow$ | $\mathcal{L}_{\mathbf{ROI}} \cdot 10^3 \downarrow$ | $\mathbf{RPFD} \downarrow$ | $\mathbf{CD_{UL1}[mm]} \downarrow$ | $J(M_{\mathbf{P}}, M_{\mathbf{GT}}) \uparrow$ |
|---|---|---|---|---|---|
| Validation set (13 objects) | 3.93 | 3.51 | 0.698 | 6.229 | 0.836 |
| Foam cylinder | 4.78 | 2.38 | 0.652 | 8.659 | 0.794 |
| Plush volleyball | 2.78 | 3.16 | 0.182 | 6.260 | 0.899 |
| Sponge | 9.34 | 3.03 | 0.603 | 8.314 | 0.731 |
| Toilet paper roll | 15.43 | 10.67 | 0.387 | 7.546 | 0.754 |

Table 14: Generalization results for 4 unseen objects using images and robot data as input.

| Input | $\mathcal{L}_{\mathbf{PFD}} \cdot 10^3 \downarrow$ | $\mathcal{L}_{\mathbf{ROI}} \cdot 10^3 \downarrow$ | $\mathbf{RPFD} \downarrow$ | $\mathbf{CD_{UL1}[mm]} \downarrow$ | $J(M_{\mathbf{P}}, M_{\mathbf{GT}}) \uparrow$ |
|---|---|---|---|---|---|
| Validation set (13 objects) | 5.07 | 4.36 | 0.737 | 6.840 | 0.814 |
| Foam cylinder | 8.13 | 5.42 | 1.206 | 10.820 | 0.738 |
| Plush volleyball | 8.83 | 8.19 | 0.452 | 9.561 | 0.829 |
| Sponge | 15.90 | 8.409 | 1.117 | 8.410 | 0.640 |
| Toilet paper roll | 37.31 | 40.45 | 0.650 | 9.952 | 0.667 |

