# OpenReview forum: "PokeFlex: A Real-World Dataset of Deformable Objects for Robotics"
_ICLR.cc/2025/Conference — Submitted to ICLR 2025_

### Official Review · Reviewer_BD6s · 2024-11-02

**Soundness:** 2
**Presentation:** 3
**Contribution:** 2
**Rating:** 3
**Confidence:** 4

**Summary:**

The paper mainly introduce a deformable objects realworld datasets featuring real-world paired and annotated multimodal data. The datasets can be mainly used for online 3D mesh reconstruction. The dataset has different data modalities and is primarily collected using a professional capture system, with the collection process also incorporating cameras such as Kinect and RealSense, and simultaneously synchronizing with the robotics system. The paper further shows the application of this datasets on mesh reconstruction and has comprehensive evaluation of learning-base reconstruction.

**Strengths:**

+ RealWorld dataset: The most prominent feature of this paper is that it presents a real-world dataset featuring real-world meshes. Benchmarks for real-world meshes in robotics are relatively scarce. Additionally, the paper demonstrates good reproducibility, providing detailed 3D printing information in the appendix.
+ Multimodal and Rich: Furthermore, this dataset is a multimodal dataset that, in addition to using high-cost acquisition systems, incorporates the commonly used RealSense camera as an eye-in-hand sensor and the Kinect as an eye-on-base sensor. The settings align well with robotic tasks and visual tasks within the field.
+ Comprehensive evaluation on reconstruction task: Finally, the paper conducts detailed experiments on reconstruction tasks, including the use of RGB data and multimodal data.

**Weaknesses:**

+ The purpose of the dataset: One major issue with this paper is that it seems there are few other application tasks beyond online mesh reconstruction for which the dataset can be utilized.  Although online mesh reconstruction is an interesting computer vision task in itself, it appears that there are few practical applications for online mesh reconstruction. Therefore, this work seems to fall short of expectations in various fields. I will elaborate on this point from three different areas.
     + robotics: In robotics, especially in the field of deformable object manipulation, meshes are rarely used; point clouds are more commonly employed[2][3][4]. The main challenges in this area are often related to occlusion and pose estimation issues. For example, Garment Tracking [1] addresses the problem of garment point cloud completion, but this work does not seem to tackle occlusion problems, thus reducing the significance of online mesh reconstruction in this context.
     + CV: This task has limited difficulty in the computer vision field, primarily because the occlusion problem is less significant and the deformation of plush toys is minimal compared to garments. For instance, in dropping tasks, there appear to be only translational and rotational changes. Real-time 6D pose detection has already been extensively studied, so this part of mesh reconstruction can be easily addressed using mesh + pose estimation techniques.
+ Number of object: The scale of 18 objects is small, and each object providing only two trajectories (poking and dropping) results in a limited data size. This makes it difficult to serve as a large-scale evaluation and can only be used for real-world fine-tuning. Papers in similar categories  such as garmentlab[2] have scanned nearly 100 objects, including at least 20 complex plush toys.
+ Object Selection: The deformable objects chosen for this paper are somewhat too simple. There are many more complex plush toys, such as teddy bears, which could also be used for these tasks and would better reflect real-world scenarios. These more complex objects often involve occlusion issues, making the work more meaningful. Most of the objects selected in this paper are simple shapes like cubes or cylinders, which limits the scope and applicability of the task.
+ Action Selection: The actions chosen, poking and dropping, are overly simplistic and lack practical value, especially in the robotics domain. The paper could benefit from including more complex actions, such as pick-and-place, which would add significant practical value and better reflect real-world robotic tasks.
+ Simulation Evaluation: The authors mention in the paper that existing simulation methods require real-world fine-tuning. However, simulations actually perform very well in tasks such as dropping and poking, for example, using the FEM algorithm in IsaacSim. The authors need to include validation within simulators to verify whether the meshes scanned in the paper can be used for simulation and to conduct a thorough analysis of the differences between current simulation algorithms and real-world conditions. This would strengthen the setting of the paper.
+ Action Protocol: The action protocol in this paper is vague. I recommend that the authors follow established protocols from datasets like YCB or GarmentLab[2] to refine and standardize their action protocols. This would enhance the clarity and reproducibility of the experiments.

[1] GarmentTracking: https://github.com/xiaoxiaoxh/GarmentTracking

[2] GarmentLab: https://garmentlab.github.io/

[3] UniGarmentManip: https://warshallrho.github.io/unigarmentmanip/

**Questions:**

+ Groundtruth mesh: If my understanding is correct, the ground truth mesh is obtained using an MVS (Multi-View Stereo) system. The paper's discussion of this point is somewhat unclear.
+ Multimodal: In typical robotic tasks, the position and action of the end-effector are also crucial. Here, the authors use a transparent stick for poking, but the dataset should incorporate the point cloud or mesh of the stick into the trajectory. This can be simply achieved by combining the robot's position with the mesh of the stick. Without this integration, it is difficult to effectively utilize the robot action information provided in the dataset.
+ Error of UR force and torque: The force and torque errors of the UR robot are significantly higher compared to other robots, such as the Franka Emika robot, as noted in the official product sheets.[1] The authors should discuss the impact of these errors on the experiments and results presented in the paper.

[1] https://assets.ctfassets.net/oxcgtdo88e20/4zQbvCXiDH2RzaaiiuwWci/ca242c85669825e8c25ff3467f8774da/UR5e_Product_Factsheet_EN_WEB.pdf

**Details Of Ethics Concerns:**

No ethnic review needed.

---

> ### Author Response · Authors · 2024-11-21
> **Part 1 - Response to Reviewer BD6s**
>
> - **W1 - Other applications of the dataset and partial observations**:   Regarding other applications see **general response 2**. Moreover, we note that the models presented by Pokeflex deal with partial observations. The sequences of images from a single viewpoint, or the point cloud data from the Kinect sensor are indeed partial observations, from which the presented models are able to recover deformed 3D meshes given a template mesh.
>
>
> - **W2 - Large scale evaluation and Related work**:  The paper never claimed to be a tool for large-scale evaluation. PokeFlex is a smaller, high-quality dataset that can indeed be used for fine-tuning as the reviewer suggests.
> Regarding the related work presented in GarmentLab, we remark that the paper was uploaded to Arxiv, on the 2nd of November 2024, one month after the ICLR deadline. As such it was not possible for us to include that reference in our related work. The objects in such a dataset fall in the category of static scans, and as we mentioned in L86-L89, statics scans are unsuitable for capturing the temporal dynamics of objects undergoing deformations.  We further note, that, to the date of publishing this response, GarmentLab only provides 3 real-world static scans of plush toys, as detailed in their appendix and link in their GitHub repo (https://drive.google.com/drive/folders/1CqJILIK8VQ-RCuLa_aFN-WtYTbovpFga  Assets/ReaWorld). In contrast, we provide multimodal time-sequence recordings of 17 volumetric deformable objects under active deformation. The latest version of our paper now includes Garmentlab in the related work.
>
> - **W3 - Number of object and object types**: . The dataset consists of objects with different levels of difficulty, going from simple geometric shapes (primitives) to complex 3D geometries like the octopus, the turtle, the Stanford bunny, and the 3D-printed heart. The decision to include simpler geometries allows the development of new techniques methodically, i.e., identifying material parameters for the foam dice should be easier than for more complex objects like the Stanford bunny. Finally, while we recognize that the Teddy bear from GarmentLab is a complex object, we remark that Pokefex also includes complex plush toys such as the octopus and the turtle.
>
> - **W4 - Action selection**:  Extending our comments in the **general response 4**, we remark that PokeFlex consists mostly of poking sequences and we argue that the dropping sequences are also quite relevant.  Consider the work in [1] that evaluates a method to learn neural constitutive laws on two volumetric deformable objects during dropping. Taking the octopus as an example, we agree that the object does not deform significantly during free fall, however, the object also exhibits large deformations as soon as it makes contact with the floor, as can be seen in the last frames of the sample sequence provided.  As such, further extensions of works like [1] can easily benefit from the dropping sequences of PokeFlex.
>   - [1] “Pingchuan Ma et al, Learning Neural Constitutive Laws From Motion Observations for Generalizable PDE Dynamics”
>
>
> - **W5 - Positioning wrt to deformable object simulation and sim / real differences**:  While the deformable object simulation has been significantly improved on IsaacSim, the sim-to-real gap is still large. Not even the official developers of IsaacSim can comment on the general “sim-to-real” gap for deformable objects right now. See discussion on IsaacLab repo.  https://github.com/isaac-sim/IsaacLab/issues/723 .
> We are currently in contact with several simulation teams from both academia and industry to evaluate potential ways in which our dataset can help tune their simulators. A thorough analysis of volumetric deformable object simulation compared with the real-world behavior captured by our dataset is then out of the scope of this submission and left as future work.
>
> - **W6 - Action protocols**. **See W4**.

---

> > ### Author Response · Authors · 2024-11-25
> > **Additional concerns and reevaluating score.**
> >
> > We hope that you found our new experiments and additional information valuable. As the end of the rebuttal is approaching, we would like to ask if you have any remaining concerns. If not, please consider raising your score accordingly.

---

> > > ### Comment · Reviewer_BD6s · 2024-12-02
> > >
> > > Thank you for your response. During the rebuttal period, authors have dedicated significant effort to engaging with the reviewers' concerns, which is good.
> > >
> > > However, I believe more time should have been spent on conducting additional experiments. Requesting supplementary experiments, such as attempting to simulate a mesh within a simulator and incorporating the manipulator's gt-mesh into the data, does not seem unreasonable. Overall, this rebuttal has largely failed to address my concerns; questions still remain, including whether gt-mesh or recovered mesh can indeed be used for simulation purposes.
> > >
> > > I hope that in future revisions, the authors will actively incorporate the reviewers' suggestions to improve the paper. This is the purpose of the discussion process. I will maintain my score.

---

> ### Author Response · Authors · 2024-11-21
> **Part 2 - Response to Reviewer BD6s**
>
> - **Q1 - Origin of ground truth mesh**. Yes. As mentioned in L091   -094, "we leverage a professional multi-view volumetric capture system (MVS) that allows capturing detailed 360° mesh reconstructions of deformable objects over time
> (Collet et al., 2015)". To further clarify this, we updated the manuscript as follows:
>
>       "we leverage a professional multi-view volumetric capture system (MVS) that allows capturing detailed 360° mesh reconstructions of deformable objects over time (Collet et al., 2015), **which we use as ground truth meshes**."
>
>  - **Q2 - Use of transparent stick for poking**: Currently the robot data includes the force in the contact point and the position of the contact point. The fields in the robot data will be extended to include the origin of the tool, and the kinematic configuration of the robot per time step. With that information, any user of the dataset can easily generate a point cloud or a mesh for the poking stick (length 192 mm, radius 10 mm), if needed. However, we note that the transparent stick does not appear in the raw depth images nor in the raw point clouds recorded by the Azure Kinect cameras. For clarity, we updated the manuscript with the following description in L188-190:
>
>         “The dataset also provides the CAD model for the mounting tool, which holds two RealSense cameras and a 192 mm long acrylic stick with a radius of 10 mm.”
>
> - **Q3 - Accuracy of force measurements:** : Prior to the data collection we validated the accuracy of the force measurements of the UR5e robot using a force Sensor (Bota MinieOne Pro) which has a 40mN resolution in the vertical/poking direction. The results of such a test can be seen in the Plot 1 below.  Note that the error of the force sensing from the UR5e only becomes large when applying large forces. Otherwise, the UR5e robot demonstrates sufficiently reliable force readings.
>   - Plot 1: https://github.com/anonymized-pokeflex-dataset/anonymized-pokeflex-dataset.github.io/blob/main/static/docs/PokeFlex_ICLR_2025_force_accuracy.pdf

---

> > ### Author Response · Authors · 2024-11-25
> > **Q2. [Update]**
> >
> > - Regarding the additional information required to render the acrylic stick, we have updated the `robot_data.json` file to include the joint positions of the UR5e robot per time step as well as the fields **T_WT**, **T_WE** and **T_WB**, which indicate the position and orientation of the acrylic stick, the end effector and the base of the robot, in the world coordinate system, respectively. The `robot_data.json` file has been updated in the zip file `Dice_T1_sample.zip` and can also be found on the following link:
> >   - https://github.com/anonymized-pokeflex-dataset/anonymized-pokeflex-dataset.github.io/blob/main/static/misc/robot_data.json

---

### Official Review · Reviewer_gn5G · 2024-11-02

**Soundness:** 3
**Presentation:** 2
**Contribution:** 3
**Rating:** 5
**Confidence:** 3

**Summary:**

The paper presents PokeFlex, a dataset that features real-world paired and annotated multimodal data, including 3D textured meshes, point clouds, RGB images, and depth maps. This dataset aims to enhance various downstream tasks and to demonstrate its effectiveness in online 3D mesh reconstruction.
PokeFlex consists of 18 deformable objects with varying stiffness and shapes. Deformations are generated by dropping these objects onto a surface or by interacting with them using a robot arm, with corresponding interaction forces and torques recorded. Additionally, the authors introduce a novel deep learning model to tackle 3D mesh reconstruction using diverse input data types.

**Strengths:**

The PokeFlex dataset stands out as an original contribution to the field. It features a diverse set of objects, including 3D-printed and everyday objects. This variety enriches the dataset and makes it applicable to a wide range of research scenarios. The inclusion of different types of input data—such as 3D textured meshes, point clouds, RGB images, and depth maps—enhances its utility for various downstream tasks.
The paper clearly conveys the significant effort involved in acquiring this dataset, highlighting the meticulous process behind its compilation. Additionally, the introduction of an innovative deep learning model to address 3D mesh reconstruction challenges is particularly noteworthy, showcasing the authors' commitment to advancing the field. Furthermore, the paper is well-structured and easily readable, making it accessible to a broad audience.

**Weaknesses:**

The paper presents results for only a limited subset of the 18 objects in the PokeFlex dataset—showing performance for just one object in some instances, five in others, and graphs for three. To better assess the dataset's overall effectiveness, it would be beneficial to provide results for all objects. This comprehensive analysis would establish a solid baseline for future comparisons.
Moreover, the choice of a transparent acrylic stick used to poke was not adequately justified or discussed. The implications of this choice on RGB and depth data need more attention, particularly regarding how reflection and refraction may distort images and how time-of-flight depth readings may be inaccurate around the stick.
To further evaluate the dataset's effectiveness, it would be helpful to have a benchmark model trained on the dataset itself to determine its robustness and construction quality. While the paper mentions the "dropping" method, the experiments reported focus solely on "poking." Including results from the dropping method would provide additional insights into the dataset's applications.
Additionally, the model that utilizes multiple data modalities only explored combinations of images and robot data. It would be interesting to see how other combinations of data types perform.
Moreover, in Equation 3, the threshold epsilon is not defined, nor is there clarity on how values like 0.2 or 0.5 are determined. Addressing these points would enhance the clarity and rigor of the research.
Additionally, several citations in the references appear to be incorrect or improperly formatted, specifically the third, thirteenth, fourteenth, eighteenth, and twenty-ninth citations.
Another point concerns object licensing: one of the objects (Stylized pizza slice) in the dataset is reportedly subject to a non-distributable license, yet this restriction is not noted. This oversight may lead to potential misuse or distribution issues.
Finally, neither the full dataset nor the trained deep learning model is currently available, which limits reproducibility. The sample dataset provided in the supplementary materials includes only one object, a dice, rather than a representative subset or the entire collection. Making the full dataset and model accessible would not only improve transparency but also enable a more comprehensive evaluation by the research community.

**Questions:**

1) In Equation 3, the value of epsilon is not specified. What value does epsilon take to define that the point p_i​ is close enough to the contact point t?
2) In Equation 3, how was the value of 0.2 chosen, and what is the reasoning behind this choice?
3) Why is the ROI loss weighted by 0.5 in the total loss calculation?
4) In Section 4.2, it is mentioned that the train-validation split was generated by randomly selecting one sequence per object. Are the sequences consistent across all objects? For the multi-object training split (Section "Learning from different data modalities"), how was the split handled? Did you consider a sequence in the validation set for one object that fell into the training set for another?
5) For the single-object training described in the section "Learning from RGB images of different cameras", could you provide results for additional objects? Results from only one object do not yield a statistically significant sample for generalizing findings.
6) For Figure 11, could you include data for additional objects (potentially in the supplementary materials)? In this case, what is the training setup—is the dataset composed of 20k samples, or just the 8.4k active paired poking frames? If the dataset used was the full 20k samples, how was it balanced to avoid an overrepresentation of frames without poking compared to those with poking?
7) Still regarding Figure 11, the x-axis represents distance values—how many samples are averaged at each value? Are there more samples at 0.1 than at 0.4? If so, how is this imbalance managed during training and validation?
8) Were environmental conditions, such as lighting, explicitly considered during dataset acquisition? Was any analysis performed to evaluate how variations in lighting or surrounding conditions might influence the networks used?

**Details Of Ethics Concerns:**

It is important to specify whether the objects used in the dataset are under open licenses. For instance, as noted earlier, the pizza model appears to have a license that restricts redistribution. Clarifying the licensing status of each object in the dataset would help ensure compliance with legal guidelines and provide transparency to other researchers regarding any limitations on dataset sharing and reuse.

---

> ### Author Response · Authors · 2024-11-21
> **Part 1 - Response to Reviewer gn5G**
>
> We have extracted a total of 8 weaknesses from your text. In the following we would like to address these and answer your questions:
>
> - **W1 - Results show only a subset of the images (Line 1-4)**: We have updated our results to consistently include all objects in the experiments **(See tables below and general response 3).** Additionally, we provide detailed results for all objects for several modalities (**see Appendix A.5**).
>
>   - **Appendix A.5**: https://github.com/anonymized-pokeflex-dataset/anonymized-pokeflex-dataset.github.io/blob/main/static/docs/PokeFlex_ICLR_2025_AppendixA5.pdf
>
> - Table 3:
>
>     | Input              | **$\mathcal{L}_{\text{PFD}} \cdot 10^3 \downarrow$** | **$\mathcal{L}_{\text{ROI}} \cdot 10^3 \downarrow$** | **$\text{RPFD} \downarrow$** | **$\text{CD}_{\text{UL1}} \text{[mm]} \downarrow$** | **$J(\mathcal{M}_{\text{P}}$**, **$\mathcal{M}_{\text{GT}}) \uparrow$** |
>     |--------------------|------------------------------------------------------|---------------------------------------------------|------------------------------|-----------------------------------------------------|-------------------------------------------------------------------|
>     | Volucam            | **6.69**                                             | 8.38                                              | 0.689                        | **7.433**                                           | 0.799                                                             |
>     | RealSense          | 7.91                                                 | **7.37**                                          | **0.693**                    | 7.675                                               | **0.806**                                                         |
>     | Kinect             | 11.20                                                | 9.79                                              | 0.839                        | 8.505                                               | 0.767                                                             |
>     | Rendered           | 12.97                                                | 12.28                                             | 0.826                        | 8.761                                               | 0.761                                                             |
>     | Brighter RealSense | 14.57                                                | 13.25                                             | 0.853                        | 9.126                                               | 0.754                                                             |
>     | Darker RealSense   | 13.89                                                | 11.74                                             | 0.957                        | 9.831                                               | 0.736                                                             |
>
> - Table 4:
> | Input                                    | **$\mathcal{L}_{\text{PFD}} \cdot 10^3 \downarrow$** | **$\mathcal{L}_{\text{ROI}} \cdot 10^3 \downarrow$** | **$\text{RPFD} \downarrow$** | **$\text{CD}_{\text{UL1}} \text{[mm]} \downarrow$** | **$J(\mathcal{M}_{\text{P}}$**, **$\mathcal{M}_{\text{GT}}) \uparrow$** |
> |------------------------------------------|------------------------------------------------------|------------------------------------------------------|------------------------------|-----------------------------------------------------|-------------------------------------------------------------------------|
> | Images                                   | 6.69                                                 | 8.38                                                 | 0.698                        | 7.43                                                | 0.799                                                                   |
> | Robot data                               | 7.43                                                 | 5.80                                                 | 0.847                        | 8.01                                                | 0.785                                                                   |
> | Images + robot data                      | 5.39                                                 | 5.10                                                 | 0.594                        | 6.64                                                | 0.821                                                                   |
> | Dense synthetic point clouds (5k points) | **4.76**                                             | **4.92**                                             | **0.569**                    | **6.33**                                            | **0.831**                                                               |

---

> > ### Comment · Reviewer_gn5G · 2024-11-25
> > **W1**
> >
> > I appreciate your effort in providing training for all objects, as this is crucial for ensuring a fair analysis of the results. However, I observed a significant drop in performance in the new experiments. It is mandatory to discuss this reversal of results and provide an explanation for why it occurred. Additionally, the discussion of the results in lines 472–474 contains inaccuracies. Please revise these sections, as the conclusions drawn there are no longer valid.

---

> > > ### Author Response · Authors · 2024-11-25
> > > **Response part 1**
> > >
> > > - The decrease in performance in the new results is due to certain objects for which the deformation predictions are less accurate. As detailed in the tables showing per-object metrics (Appendix A.5), objects like the toilet paper roll, beanbag, memory foam, and the 3D-printed bunny exhibit significantly worse performance compared to others. This contributes to a higher average error across all objects.
> > >
> > > - We have updated the sections in the discussion “Learning from RGB images from different cameras” and “Multi-object mesh reconstruction from different modalities” accordingly. Additionally, we now provide results for all experiments, including models trained on all objects for sparse point clouds, Kinect point clouds, and Kinect point clouds combined with robot data (**see the updated version of Table 4 below**).
> > >
> > > Table 4:
> > > | Input                                      | **$\mathcal{L}_{\text{PFD}} \cdot 10^3 \downarrow$** | **$\mathcal{L}_{\text{ROI}} \cdot 10^3 \downarrow$** | **$\text{RPFD} \downarrow$** | **$\text{CD}_{\text{UL1}} \text{[mm]} \downarrow$** | **$J(\mathcal{M}_{\text{P}}$**, **$\mathcal{M}_{\text{GT}}) \uparrow$** |
> > > |--------------------------------------------|------------------------------------------------------|------------------------------------------------------|------------------------------|-----------------------------------------------------|-------------------------------------------------------------------------|
> > > | Images                                     | 6.69                                                 | 8.38                                                 | 0.698                        | 7.433                                               | 0.799                                                                   |
> > > | Robot data                                 | 7.43                                                 | 5.80                                                 | 0.847                        | 8.014                                               | 0.785                                                                   |
> > > | Images + robot data                        | 5.39                                                 | 5.10                                                 | 0.594                        | 6.642                                               | 0.821                                                                   |
> > > | Dense synthetic point clouds (5k points)   | **4.76**                                             | **4.92**                                             | 0.569                        | **6.338**                                           | **0.831**                                                               |
> > > | Sparse synthetic point clouds (100 points) | 6.14                                                 | 5.61                                                 | 0.577                        | 6.569                                               | 0.815                                                                   |
> > > | Kinect point clouds                        | 6.17                                                 | 6.56                                                 | 0.592                        | 6.619                                               | 0.807                                                                   |
> > > | Kinect point clouds + robot data           | 6.86                                                 | 6.18                                                 | **0.539**                    | 6.613                                               | 0.817                                                                   |

---

> > > ### Author Response · Authors · 2024-11-25
> > > **Additional concerns and reevaluating score.**
> > >
> > > We appreciate your response and your detailed feedback. We’re glad that you found the extended experiments valuable. Please, let us know if you have any additional concerns. Otherwise, we kindly ask you to consider raising your score accordingly.

---

> > > > ### Comment · Reviewer_gn5G · 2024-11-26
> > > > **Open Issues**
> > > >
> > > > 1) Lack of Experiments for Dropping Data
> > > > The authors chose not to provide experimental results or analysis for the dropping data, leaving its inclusion in the study unsubstantiated.
> > > > Recommendation: Conduct basic experiments using the dropping data to justify its relevance and integration into PokeFleX.
> > > >
> > > > 2) Risk of Overfitting in Train-Validation Splits
> > > > While the authors describe data diversification strategies, they do not adequately address how these strategies mitigate overfitting risks or avoid potential issues with improper train-validation splits.
> > > > For example, consider a scenario involving a video sequence dataset (different from yours, but for illustrative purposes only). If the dataset is split randomly into training and validation sets, the i-th frame may be included in the validation set while the (i+1)-th frame falls into the training set. The minimal differences between consecutive frames create an overlap, leading to a flawed validation process where the model is evaluated on data it effectively "saw" during training. In your case, does the selection of sequences encounter the issue illustrated in the example?
> > > > Recommendation: Include ablation studies or perform cross-validation to better understand and address the risk of overfitting and ensure proper splitting practices.
> > > >
> > > > 3) Performance Changes in Figure 11 and Table 3
> > > > In Table 3, there is no discussion regarding the reversal of results. "Rendered" input was always the best, while now it is not.
> > > > In Figure 11, the authors attribute performance changes to an increased representation of spherical objects. However, this explanation lacks supporting data and fails to address the specific question: Why does the new figure differ from the older one? The provided explanation regarding differences in deformation distributions is too general.
> > > > Recommendation: Clarify the observed performance differences with detailed analysis and supporting data, addressing the specific concerns about these figures.
> > > >
> > > > 4) Benchmark Models for Dataset Evaluation
> > > > The suggestion to assess the dataset’s effectiveness using state-of-the-art (SOTA) benchmark models was not addressed.
> > > > Recommendation: Incorporate benchmark models to evaluate the dataset and validate its utility in a broader context.
> > > >
> > > > The paper would benefit from addressing these points through experiments, ablation studies, and providing more detailed analyses to substantiate the claims and improve soundness. Without these elements, I do not feel confident in raising the rating.

---

> > > > > ### Author Response · Authors · 2024-11-27
> > > > > **Response.**
> > > > >
> > > > > - **R1**: We argue that the inclusion of the dropping sequences is highly justified by the many potential applications of the dataset, as exemplified by the non-exhaustive list of applications in **general response 2**. Such applications are valid for both poking and dropping.
> > > > >
> > > > >   An additional reason to include the dropping sequences in the dataset is to showcase the feasibility of reconstructing meshes deforming using strategies different than poking.  Furthermore, as mentioned in previous responses, dropping experiments are easier to reproduce (See Response to W3 in  Part 2 https://openreview.net/forum?id=XwibrZ9MHG&noteId=YMezrO6i6F ), as any user can drop objects without the need of a robot.
> > > > >
> > > > >   If despite our arguments, the reviewer considers that the inclusion of the dropping sequences is not justified, we are open to excluding such sequences from the dataset, as they were intended to be the cherry on top of PokeFlex.  We remark that the dropping sequences constitute only 16% of the total number of frames in our dataset (L344-346.), and therefore even with the exclusion of such sequences, PokeFlex still stands out as an original contribution to the field, as remarked by the reviewer themself  (https://openreview.net/forum?id=XwibrZ9MHG&noteId=wB60wOvDyp ).
> > > > >
> > > > >   Nonetheless, to reassure the reviewer about the importance of such sequences, we would like to mention that we are currently conducting experiments using the dropping data in simulation to identify material parameters. However, we consider this as future work and our current results are preliminary and not mature enough to be shared.
> > > > >
> > > > > - **R2**: We are afraid that there is a misunderstanding in how we are performing the split between training and validation samples. For each object, we have several recordings that consist of 150-180 frames each. In each of those recordings,  the robot performs 4-5 vertical poking movements to deform the objects at different positions. Between recordings, the objects are randomly rotated and repositioned to increase diversity in the collected data. As a validation set,  a complete recording of 150-180 frames is chosen for each object. Therefore it is impossible that frame i is in the training set and frame i+1 of the same recording is in the validation set.
> > > > >
> > > > > - **R3**: In general, it is difficult to expect the results of all experiments to remain consistent when training for multiple objects at the same time, especially considering that some of the newly included objects for training have more complex geometries and internal structures (like the 3D-printed objects),  which lead to a more diverse set of deformation behaviors.
> > > > >
> > > > >    Regarding **Table 3**, we note that our previous results were based on models that were trained on only one object (the foam dice). We hypothesize that when training with multiple objects from image-only data, the models trained with images from the Volucam and the  Realsense cameras perform better than when using images rendered from the reconstructed meshes, because of the additional spatial information that is implicitly encoded in such images, as they also contain pixels that reflect the robot and its motion, which is not the case for the rendered images.
> > > > >
> > > > >    Regarding **Figure 11**, we remark that when using all objects for training,  the task of image-based reconstruction becomes more challenging and the mean performance over all objects naturally decreases.  Moreover, to further support our arguments explaining why the plush moon is now the object where the best performance is achieved, we note that a better-than-average performance in terms of Jaccard index and RPFD  is obtained for all the sphere-like objects reported in **Table 10 - Appendix A.5** (plush moon, plush volleyball, and foam half sphere).
> > > > >
> > > > >    Regarding the differences in the deformation distributions across various objects, we respectfully disagree with the assessment that our explanation is overly general. However, to offer further insights into why the distributions are so diverse, we remark that apart from having diverse material properties, the objects in PokeFlex also have complex geometries with very different internal structures.  We want to clarify that even though we used similar poking strategies (we always poked from the top), an additional source of randomness for the deformation comes from the randomization of the poking depth and the placement of the object, which leads to contacts in different points. With such randomness and such a diversity of objects, it’s difficult to expect any patterns from the deformation distributions.
> > > > >
> > > > > - **R4**: As none of the previous comments from the reviewer contain any references to papers, we would be thankful if the reviewer could clarify which are the suggested SOTA methods for multimodal online 3D-mesh reconstruction of deformable objects that we should use to position PokeFlex?

---

> ### Author Response · Authors · 2024-11-21
> **Part 2 - Response to Reviewer gn5G**
>
> - **W2 - Reason for using an acrylic stick and its implications on ToF-depth readings (Line 4-7)**: The acrylic stick was chosen to minimize occlusions on the object, leading to better reconstruction quality. In a previous setup, we used a Robotiq gripper for poking, which resulted in lower-quality reconstructions and the loss of texture in the deformed region, due to occlusions. We do not observe any impact on the Azure Kinect’s depth readings caused by the stick, as it does not appear in any point cloud visualizations.
>
> - **W3 - No experiments for dropping (Line 8-10)**: While we currently lack experiments using the dropping data, we believe these sequences are valuable due to their higher reproducibility compared to poking (Any user of the dataset can drop objects without the need of a robot). Additionally, they hold potential for fine-tuning simulators and extending existing works, such as [1].
>
>     - [1] “Pingchuan Ma et al., Learning Neural Constitutive Laws From Motion Observations for Generalizable PDE Dynamics”
>
> - **W4 - Combination of modalities only for images and robot data (Line 10-12)**: We are actively running new experiments that include the combination of point clouds and robot data. These results will be included during the rebuttal period.
>
> - **W5 - Choice of thresholds (Line 12-14)**: The thresholds were determined heuristically based on experimental observations.
>
> - **W6 - Incorrect citation format (Line 14-15)**: Thank you for pointing this out. We have corrected the citation formatting in the updated version of the paper.
>
> - **W7 - Non-distributable license for the pizza object (Line 15-17)**: We are in contact with the authors of Pizza model to address this issue. For now, we have removed the pizza from the dataset and all related experiments for the rebuttal period. For the camera-ready submission, we plan to use our own pizza model. We have already 3D printed a first version (**See video 1**).
>   - **Video 1**: https://github.com/anonymized-pokeflex-dataset/anonymized-pokeflex-dataset.github.io/blob/main/static/videos/new_3d_printed_pizza.mp4
>
>  - **W8 - Dataset not fully released (Line 17-21)**: The full dataset will be released by the camera-ready deadline (upon acceptance), as is common practice in other major conferences. The dataset is >500GB, and while we have the capacity to release such large datasets, current restrictions prevent us from releasing it without violating the double-blind policy. As stated in the paper, we are committed to making the entire dataset publicly available.
>
> - **Q1 - Choice of epsilon**: The epsilon value was set to 0.4, as this provided a good overlap between the region of interest and the poking region of many objects.
>
> - **Q2 - Choice of values in Equation 3**: The condition in Equation 3 ensures that the bottom surface of the object is not included in the region of interest, which can occur for thin objects like the sponge. As stated in L359, we scale the meshes to a unit cube, which justifies the value of 0.2 aiming to exclude the bottom surface while including the deformed region.
>
> - **Q3 - Choice of weight for LRoi**: The weight for LRoi was selected heuristically to balance its contribution to the overall loss function.
>
> - **Q4 - Train-validation split**: For multi-object training, we randomly selected one poking sequence per object for validation, with all other sequences used for training. All evaluations are based on the extracted validation sequences.
>
> - **Q5 - Experiments for all objects**: Additional experiments now include all objects (**see general response 3**).
>
> - **Q6 - Including more objects in Figure 11 and training setup**: We have generated equal graphs as Figure 11 for all objects, along with a histogram showing the sample distribution for each object (**see Appendix A.6**). For the training setup, the dataset contains only samples in deformed configurations.
>     - **Appendix A.6**: https://github.com/anonymized-pokeflex-dataset/anonymized-pokeflex-dataset.github.io/blob/main/static/docs/PokeFlex_ICLR_2025_AppendixA6.pdf
>
> - **Q7 - Distribution of samples**: We now provide insights into the distribution of the data using histograms (**see Appendix A.6**)
>     - **Appendix A.6**: https://github.com/anonymized-pokeflex-dataset/anonymized-pokeflex-dataset.github.io/blob/main/static/docs/PokeFlex_ICLR_2025_AppendixA6.pdf
>
> - **Q8 - Evaluation under different lighting conditions**: Environmental conditions were not explicitly varied during data acquisition since consistent lighting is integral to the MVS system for proper reconstruction. However, we introduce new experiments to evaluate the model’s robustness to modified lighting (**see Table 3 in W1**). Lighting modifications were achieved by adding or subtracting a constant value, and introducing random noise to all image channels. We applied the modifications to the RealSense images and evaluated on the model that was trained on the ground truth RealSense images.

---

> > ### Comment · Reviewer_gn5G · 2024-11-25
> > **Part2**
> >
> > W5, Q1,Q2, Q3) I believe an ablation analysis would greatly enhance these values by offering a stronger justification and a clearer rationale.
> > Q4) Selecting one sequence per object for validation does introduce a potential risk of overfitting, especially if the sequences within an object are highly similar or exhibit minimal variability. This could result in the model appearing to perform well on the validation set without demonstrating strong generalization. Which strategies did you use to avoid it?
> >
> > Q6) Figure 11: why does it change with worse performances? In the previous version, the best was the dice, while in this one it is the push moon.
> >
> > Q7) I would like to understand the histograms in Figures 14-19: why are the distributions so different? For example, I am referring to the 3D printer cylinder or the Toilet Paper Roll. Please explain the motivations

---

> > > ### Author Response · Authors · 2024-11-25
> > > **Response part 2**
> > >
> > > - We plan to conduct a hyperparameter search for all heuristic set parameters to provide a more robust justification for our chosen values, for the camera-ready version. Regarding overfitting and to maximize variability between sequences, the objects were randomly repositioned, and for asymmetric objects, the surface in contact with the bottom was alternated. Additionally, the poking trajectory was varied across takes to introduce further diversity. For image-based reconstruction, we further employed dropout in the feature reduction layers following the DinoV2 feature extraction to reduce overfitting and improve generalization.
> > >
> > >
> > > - Including all objects in the training data increased the representation of sphere-like objects (e.g., the plush volleyball), which we believe has contributed to the improved performance of the plush moon compared to the foam dice. Nonetheless, it’s worth noting that the trained models still perform better than average for the foam dice for multiple data modalities (Tables 10, 11, and 12), supporting the initial experiments.
> > >
> > >
> > >
> > > - The histograms in Appendix A6 reveal the diversity of the material parameters of the objects used in our dataset. As such, the objects exhibit different deformation levels in response to similar poking strategies. We view this diversity in the deformation distribution as a strength of our dataset, highlighting the range of distinct deformation behaviors captured by PokeFlex.

---

### Official Review · Reviewer_RQv9 · 2024-11-02

**Soundness:** 2
**Presentation:** 3
**Contribution:** 2
**Rating:** 3
**Confidence:** 4

**Summary:**

In this paper, the authors collected a real-world multimodal dataset of 18 deformable objects. The dataset includes 3D textured meshes, point clouds, RGB images and depth maps.The authors use a volumetric capture system to build the 360degree reconstruction. It includes two types of deformation, poking and dropping.

The author further use the dataset to train neural network models for deformed mesh reconstruction based on template meshes and different modalites included in the dataset. The neural network model architecture is reasonable and the perforamnce is good on the different trajectories of the same objects.

**Strengths:**

The main advantage of this dataset is that it contains paired 3D meshes and contact forces. They further design a deformation mesh reconstruction method which takes the template mesh and force or image of point cloud sequece as input. By using the GT deformation meth for training, the trained model performs good on different trajectoris of the same objects.

**Weaknesses:**

1. According to Table2, the academic value and the difference between the submsion and PLUSH(Chenetal.,2022) should be further discussed, since it is not difficult to reconstruct the meshes from point clouds. Why do we need the mesh? What other application can it enable on top of deformation reconstruction?

2. The generalizability of the network on unseen objects is not evaluated, which may be the most valuable part of the model. If the model can only work on trained objects, the usefulness of this method is very limited.

3. The technical novelty of the network model is limited and not validated through ablation studies.

**Questions:**

1. What are the densesyntheticPoint Clouds(5kpoints) and SparsesyntheticPoint Clouds(100points)? Why are the error still large when using synthetic point clouds?

2. What is NVP? IT should be explained briefly to make the paper self-contained.

---

> ### Author Response · Authors · 2024-11-21
> **Part 1 - Response to Reviewer RQv9**
>
> - **W1 - Position wrt to PLUSH**:
>     After careful reconsideration of the PLUSH dataset, we note that such a dataset does not provide explicit force measurements in Newtons, as Pokeflex does. The PLUSH dataset reports the pose of the nozzle that applies an air stream to the different Plush toys and it provides a table (https://github.com/facebookresearch/plush_dataset/blob/main/plush_data.pdf) that reports the impact of the nozzle wrt to the distance on a weight. However, the Plush dataset does not report any method to convert such quantities to forces and they do not evaluate the accuracy of the force estimation. To clarify this, we added a new footnote “by providing air nozzle poses” to Table 2 (L117-118) in the latest version of the paper .
>
>     Furthermore, we want to emphasize that PokeFlex is not proposed as a simple point cloud to mesh reconstruction dataset.  PokeFlex provides real-world high-quality soft-body temporal deformation data using both high-end and consumer-grade capture systems, synchronized with robot interaction sensor data. Our ultimate goal is to investigate the modeling of the physical properties of soft-body objects in the context of manipulation tasks. Accurate modeling of physical properties is not possible with most datasets, as they lack annotations of the exact point of force exertion and the applied force.
>
>     As the reviewer notes, the Plush dataset is an important step towards this goal. However, in the Plush dataset, the forces applied on the objects are inevitably more global using an air stream, which causes more global deformations and hinders accurate force estimation. Such global deformation can lead to high inaccuracies, as they report in their paper:
>
>       "Larger errors are observed for objects with a thin and tall component (see the last 4 rows of the table). This error is largely caused by tracking inaccuracies of the nozzle: even slight inaccuracies can cause large errors when, for example, the neck of the dinosaur moves while the recorded air stream direction does not, or barely, touch the objects".
>
>     Regarding the second type of manipulation in the PLUSH dataset, using fishing lines, the dataset does not report tension forces of the fishing lines as the authors only incorporated such forces as soft position constraints for specific points.
>
>     Finally, we remark that Pokeflex applies local deformations at specific contact points and also deals with rigid surface contacts. While we acknowledge that the Plush dataset itself is very valuable, PokeFlex incorporates significant complementary information.
>
> - **W1 - Importance of mesh representation and potential applications**: **See general response 2.**
>
> - **W2 - Generalizability of networks on unseen objects**:
>     The main contribution of the paper is the dataset. The presented online 3D mesh reconstruction is just one of the many potential downstream applications. Nonetheless, we agree that this is an interesting experiment. As such, we include the results in the latest version of the paper (see Appendix A.7).
>
>     - Appendix A.7: https://github.com/anonymized-pokeflex-dataset/anonymized-pokeflex-dataset.github.io/blob/main/static/docs/PokeFlex_ICLR_2025_AppendixA7.pdf
>
>     For the generalizability experiments, we trained reconstruction models using sequences of synthetic point clouds or images as input. We trained on 13 objects and evaluated on 4 unseen objects, namely, the foam cylinder, the plush volleyball, the sponge, and the toilet paper roll. As a reference, Table 10 and Table 11 in Appendix A.7 also provide the performance on the validation set of 13 objects that were used during training.
>
>     We note that the model trained on point clouds tends to generalize better than the model that uses images. As expected, given the size of our dataset, the performance degrades for unseen objects. However, some level of generalization can be observed for objects like the plush volleyball that has a similar shape and size as other objects in the training set.
>
> - **W3 - Technical novelty of the network and ablation studies**: We agree that the technical novelty of the model is limited. However, we remark again that the main focus of the paper is the dataset. Regarding the ablations, we consider that the evaluations of different data modalities and their combinations for 3D mesh reconstruction serve as ablations, as they highlight the different performance results that can be obtained.

---

> ### Author Response · Authors · 2024-11-21
> **Part 2 - Response to Reviewer RQv9**
>
> - **Q1 - Synthetic point clouds**:
>     For the dense point cloud setting, we set the number of points to 5K as it is sufficiently dense to represent the geometry of the objects, while not being too big to hinder the inference speed of the model under 100Hz on an RTX4090. The sparse synthetic point cloud was set to 100 points to stress test the reconstruction method's performance since this is the regime where the quality of the mesh reconstruction starts degrading significantly.
>
>     When using synthetic point clouds, we attribute the imperfect reconstruction results to limitations in the NVP architecture that we used. Such architecture excels at capturing global deformation. However, it has a harder time reconstructing smaller / local deformations as the ones caused in the poking area.
>
>
> - **Q2 - Real-NVP architecture**: The conditional Real-NVP architecture is referenced in L240-241  and it is an architecture that given a template mesh outputs the per-vertex displacement to match a deformed mesh configuration while preserving the original mesh topology. For clarity, we added a brief description of Real-NVP architecture in L242-245 as follows:
>
>       "Real-NVP utilizes a series of conditional coupling blocks, each defined as a continuous bijective function. This continuous bijective operation ensures that the model is homeomorphic, which allows stable deformation of a template mesh while preserving its topology."

---

> ### Author Response · Authors · 2024-11-25
> **Additional concerns and reevaluating score.**
>
> We hope that you found our new experiments and additional information valuable. As the end of the rebuttal is approaching, we would like to ask if you have any remaining concerns. If not, please consider raising your score accordingly.

---

> > ### Comment · Reviewer_RQv9 · 2024-11-27
> >
> > Thank you for your response. While you highlighted that the primary value of this paper lies in the dataset, its scope is quite limited in terms of the types of objects and manipulations it covers—a point that has also been raised in other reviews. Furthermore, the paper does not sufficiently demonstrate the value of the dataset, such as its potential for fine-tuning simulations. For these reasons, I will maintain my score.

---

### Official Review · Reviewer_m6Yb · 2024-11-05

**Soundness:** 3
**Presentation:** 2
**Contribution:** 3
**Rating:** 5
**Confidence:** 4

**Summary:**

The paper introduces PokeFlex, a new dataset focused on real-world deformable objects for robotics applications. The dataset includes:

- Multimodal Data: 3D textured meshes, point clouds, RGB images, and depth maps.
- Objects: 18 deformable objects with varying stiffness and shapes, including everyday items and 3D-printed objects.
- Interaction Protocols: Deformations induced by poking with a robot arm (with force/torque data) and dropping onto a flat surface.
- Purpose: For downstream tasks like online 3D mesh reconstruction and enable applications such as real-world deployment of control methods based on mesh simulations.
- Data Acquisition: A volumetric capture system for 360 deg. reconstruction and integrates lower-cost RGB-D sensors for reproducibility.
- Baseline Models: Shows the use of PokeFlex in online 3D mesh reconstruction using different data modalities and provides evaluation criteria.

**Strengths:**

- Rich, Real-World Dataset: PokeFlex offers a comprehensive multimodal dataset with 3D meshes, point clouds, RGB images, depth maps, and force/torque measurements, captured with a high-quality volumetric system. This real-world data bridges the sim-to-real gap, enhancing applicability in deformable object manipulation.
- Reproducibility and Accessibility: Open-source 3D-printed objects and affordable sensors (Azure Kinect, Intel RealSense) make the dataset accessible and reproducible, while baseline models and evaluation metrics support benchmarking and further robotics research.

**Weaknesses:**

- Limited Object Diversity: While the dataset includes 18 objects, the diversity may still be limited compared to the vast range of deformable objects encountered in real-world applications. Notably, the dataset focuses on volumetric objects and excludes thin deformable items like cloth or cables, which are important in areas like garment handling.
- Controlled Interaction Protocols: The poking and dropping protocols are specific and may not capture the full spectrum of possible deformations in less controlled or more complex environments. Additional manipulation actions could enrich the dataset.
- Potential Bias in Data Collection: The use of a transparent acrylic stick for poking and specific dropping heights might introduce biases that limit the generalizability of the dataset to other types of interactions and objects.
- Related Work Coverage: The paper could benefit from a more thorough comparison with existing datasets and approaches in the literature, such as DOFS [1], CEPB [2], and other relevant works on deformable object datasets and manipulation. Discussing how PokeFlex complements or improves upon these datasets would strengthen the contribution.

References:
[1] Zhang, Zhen, et al. "DOFS: A Real-world 3D Deformable Object Dataset with Full Spatial Information for Dynamics Model Learning." arXiv preprint arXiv:2410.21758 (2024).
[2] Tripicchio, Paolo, Salvatore D’Avella, and Carlo Alberto Avizzano. "CEPB dataset: a photorealistic dataset to foster the research on bin picking in cluttered environments." Frontiers in Robotics and AI 11 (2024).

**Questions:**

- Q1: How do you plan to address the limitations in reconstructing fine-grained details for smaller objects in future work?
- Q2: Are there plans to extend the dataset to include thin deformable objects like cloth or ropes, which are significant in many robotic applications?
- Q3: Do you intend to incorporate more complex or varied manipulation protocols beyond poking and dropping to simulate a wider range of real-world interactions?
- Q4: Is there any intention to provide additional annotations (e.g., segmentation masks, keypoints) to facilitate other types of learning tasks?

**Details Of Ethics Concerns:**

N/A.

---

> ### Author Response · Authors · 2024-11-21
> **Response to Reviewer m6Yb**
>
> - **W1 - Other types of deformable objects and further applications**: We agree with the reviewer that cloth items and cables are very important. However, as mentioned in L521-522 , the focus of PokeFlex is on volumetric objects,  which we argue is also an important category of deformable objects and can be used as a testbed for multiple downstream applications. **See General responses 1 and 4**.
>
> - **W2/W3 - Action protocols**. **See general response 4**.
>
> - **W4 - Related work**:
>   - [1]  Regarding DOFS we remark that their work was presented as a workshop paper at CoRL 2024, which took place one month after the deadline submission for ICLR. It is not common practice for the authors of PokeFlex to cite workshop papers. However, we acknowledge that their work is relevant and it is concurrent to ours. DOFS is a pilot dataset that contains only one type of deformable object under quasi-static deformation, few camera views, and it does not report interaction forces. In contrast, PokeFlex includes multiple objects, with annotated time sequences of multiple data modalities that include interaction forces and contact points. For completeness, the latest version of our paper now includes DOFS in the related work.
>
>   - [2] CEPB primarily focuses on synthetically generated photorealistic images in static configurations, which fundamentally differ from the focus of PokeFlex. While CEPB demonstrates the utility of simulation for generating large-scale datasets, it does not include real-world dynamics of deformable objects or forces as presented in the PokeFlex dataset.
>
> - **Q1 - Dealing with fine-grained details**: As stated in ~~L459-460~~ L431 “Better fine-grained reconstruction results can be expected by rearranging the cameras in a smaller workspace”. For a smaller workspace, a smaller number of cameras should be enough to generate 360° reconstructions, which would in turn reduce the compute time required to generate the ground truth meshes.  In summary, the MVS system can enable future iterations of the dataset, focusing on fine-grained reconstruction of smaller deformable objects.
>
> - **Q2 - Plans to incorporate cloth items**: Yes, The current MVS setup already enables the reconstruction of cloth objects. (See Video 1). Future iterations/extensions of the dataset will include cloth-like objects. However, as the methods used for the reconstruction and manipulation of deformable objects are quite diverse depending on the type of deformable object  (volumetric, thin-shell (cloth), linear), we decided to focus the scope of PokeFlex on volumetric deformable objects initially.
>
>   - Video 1: https://github.com/anonymized-pokeflex-dataset/anonymized-pokeflex-dataset.github.io/blob/main/static/videos/cloth_reconstruction_example.mp4
>
> - **Q3 - Other action protocols**:  **See general response 4**.
>
> - **Q4 - Additional annotations**: Yes, we will provide a script to generate segmentation masks, which can be easily obtained using the 3D textured meshes that we provide.

---

> > ### Author Response · Authors · 2024-11-25
> > **Additional concerns and reevaluating score.**
> >
> > We hope that you found our new experiments and additional information valuable. As the end of the rebuttal is approaching, we would like to ask if you have any remaining concerns. If not, please consider raising your score accordingly.

---

### Author Response · Authors · 2024-11-21
**General response**

Dear reviewers, we are pleased that:

- Reviewer **gn5G** acknowledges PokeFlex as a novel contribution,  finds the variety of data modalities beneficial for downstream applications, and recognizes the substantial effort invested in collecting this dataset and the authors' dedication to advancing the field.
- Reviewers **m6Yb** and **BD6s** appreciate the comprehensive multimodal nature of PokeFlex and acknowledge the authors' efforts in ensuring reproducibility and accessibility.
- Reviewer **RQv9** recognizes the inclusion of 3D meshes and contact forces as a key advantage of PokeFlex.

## General concerns
In the following, we address the general concerns of the reviewers:

- **Response 1**: We decided to change the title of the paper to: "*PokeFlex: A Real-World Dataset of **Volumetric** Deformable Objects for Robotics.*", to clarify the scope of the initial version of the PokeFlex dataset wrt cloth items and ropes (**m6Yb**, **BD6s** ). This change is now reflected in the latest version of the paper.

- **Response 2**: Below, we present a non-exhaustive list of potential applications for volumetric deformable objects that  PokeFlex enables, to highlight the usefulness of our dataset and the importance of having a textured 3D mesh representation (**BD6s** , **RQv9**):

   - Material parameter identification for FEM simulators.
   - Real-world markerless deployment of traditional control methods for shape control using FEM simulators.
   - Learning dynamics models for volumetric deformable objects in a low data regime.
   - View-agnostic online 3D mesh reconstruction from static cameras, using virtual cameras and the 3D textured meshes.
   - Online 3D mesh reconstruction from moving cameras, using a moving virtual camera and the 3D textured meshes.
   - Synthetic data generation for other geometry representations such as NeRFs, Gaussian Splats, SDFs.

- **Response 3**: We extended the initial experiments to include all the objects of the dataset (excluding the Pizza object for now -*Note 1). The experiments below have already been incorporated to the latest version of the paper:
    - **Experiment 1**: Learning from RGB images of different cameras, including additional tests for lighting conditions for RGB images (Table 3) (**gn5G**).

    - **Experiment 2**: 3D mesh reconstruction of different data modalities (Table 4 -*Note 2), including new histograms that show the data distribution according to different deformation levels (Appendix A6) and a detailed breakdown of the per-object performance for models trained from sequences of images, images + robot data, and point clouds (Appendix A5) (**gn5G**).
      -  Appendix A.5: https://github.com/anonymized-pokeflex-dataset/anonymized-pokeflex-dataset.github.io/blob/main/static/docs/PokeFlex_ICLR_2025_AppendixA5.pdf
      -  Appendix A.6: https://github.com/anonymized-pokeflex-dataset/anonymized-pokeflex-dataset.github.io/blob/main/static/docs/PokeFlex_ICLR_2025_AppendixA6.pdf

    - **Experiment 3**: Generalization of 3D mesh reconstruction to unseen objects. Training on 13 objects, testing on 4 unseen objects (Appendix A7) (**RQv9**).

      -  Appendix A.7: https://github.com/anonymized-pokeflex-dataset/anonymized-pokeflex-dataset.github.io/blob/main/static/docs/PokeFlex_ICLR_2025_AppendixA7.pdf

- **Response 4**: Regarding the action protocols (poking and dropping),  we agree that more diverse actions could be useful for generalization and standardization (**m6Yb**, **BD6s**). However, achieving high levels of generalization for deformable objects purely based on real-world samples is challenging due to the high dimensionality of the state and action spaces associated with deformable objects.  We argue that the action protocols employed for PokeFlex are a solid starting point for future investigations of material parameter identification methods for FEM simulators to reduce the sim-to-real gap.  A possible approach to achieve generalization is then to leverage simulation models that are tuned with real-world data. Other manipulation strategies that we want to explore for future work are indeed pick-and-place interactions and shaking. The latter in particular should be able to excite most deformation modes. However, including such actions would lead to a decrease in the quality of the reconstructed meshes because of the occlusions generated by a parallel gripper, which goes against one of the initial goals of PokeFlex, which is to provide high-quality meshes. Considering such actions is therefore out of the scope of the first iteration of the dataset.

---

> ### Author Response · Authors · 2024-11-21
> **General response - Part 2**
>
> **Note 1**: For clarifications about the exclusion of the Pizza object see response to reviewer **gn5G**. A new custom made Pizza will be introduced later.
>
> **Note 2**: Due to compute limitations, Table 4 does not cover some of the modalities presented during the initial submission, but it will be completed during the rebuttal period, with experiments trained on all objects for Kinect point clouds, sparse synthetic point clouds, and a new combination of point clouds with force information.
>
>
>
> ## Specific concerns
> Further clarifications are provided in the responses to each reviewer, including the positioning of our work wrt concurrent related papers that became available on Arxiv several weeks after the ICLR submission deadline, as suggested by **m6Yb, BD6s**.

---

### Meta-Review · Area_Chair_JFnn · 2024-12-18

**Metareview:**

This paper introduces a dataset called PokeFlex, which contains real-world deformation data for 17 objects, captured through poking and dropping. The dataset includes various modalities of data, such as reconstructed 3D meshes, forces/torques, RGB images, and depth images. The authors describe how the data was collected and discuss potential applications of the dataset, such as deformation prediction.

All reviewers gave negative scores and raised similar concerns about the usability of the proposed dataset, particularly due to the limited number of objects, the diversity of objects, and the range of actions. Despite the experimental results and additional experiments presented in the rebuttal, the reviewers were not convinced about the dataset’s usability, and the AC concurs with the reviewers' consensus. As a result, the AC made the decision to reject the submission.

**Additional Comments On Reviewer Discussion:**

Please see the Metareview.

---

### Decision · Program_Chairs · 2025-01-22

Reject